# Ocean freshening near the end of the Mesozoic

Wiesława Radmacher ®[1] ✉, Igor Niezgodzki ®[1], Vicente Gilabert[2,3], Gregor Knorr[4], David M. Buchs ®[5,6], José A. Arz ®[3], Ignacio Arenillas ®[3], Martin A. Pearce[7,14], Jarosław Tyszka[1], Mateusz Mikołajczak[1], Osmín J. Vásquez ®[8], Sarit Ashckenazi-Polivoda ®[9,10], Sigal Abramovich[11], Mariusz Niechwedowicz ®[12] & Gunn Mangerud[13]

Paleogeographic changes have significantly shaped ocean circulation and climate dynamics throughout Earth's history. This study integrates geological proxies with climate simulations to assess how ocean gateway evolution influenced ocean salinity near the end of the Mesozoic (~66 Ma). Our modeling results demonstrate that 1) Central American Seaway shoaling reorganizes ocean currents, and 2) Arctic marine gateway restrictions, confining Arctic–Global Ocean exchange exclusively to the Greenland–Norwegian Seaway, drive Arctic Ocean surface freshening and southward outflow of buoyant, low-salinity waters. However, only the combined effect of these two factors leads to both Arctic freshening and increased water mass stratification in the Greenland–Norwegian Seaway, proto-North Atlantic, and the Western Tethys. This scenario aligns with Maastrichtian palynological, micropaleontological, and geochemical records from high- and low-latitude sites. Our findings highlight the profound impact of these latest Cretaceous paleogeographic reconfigurations in altering global salinity patterns, underscoring their role as key drivers of global climate dynamics.

Disruptions in global marine environments have played a pivotal role in shaping the history of life on Earth. The reconfigurations of landmasses and marine gateway connections are fundamental drivers of changes in global circulation patterns, biogeochemical cycles, and marine ecosystems. These long-term transformations significantly impact the salinity of surface waters in the oceans, leading to the redistribution of water masses through salinity-driven stratification. While the substantial effects of marine gateways on oceanic systems have been well-documented[1–4], and the role of global salinity distribution prior to the end of the Mesozoic is well-known[2,5–7], the more direct impact of paleogeographic reconfigurations on the global salinity budget during the latest Cretaceous remains unexplored. This is particularly surprising, given the potential of the salinity to influence surface phytoplankton biomass and productivity, which in turn affects oxygen availability within the water column, alters the ocean-atmosphere carbon dioxide equilibrium, and ultimately shapes the

[1]ING PAN – Institute of Geological Sciences, Polish Academy of Sciences, Research Center in Kraków, Senacka 1, Poland. [2]Department of Geology, Faculty of Science and Technology, University of the Basque Country (UPV/EHU), Bilbao, Spain. [3]Departamento de Ciencias de la Tierra, and Instituto Universitario de Investigación en Ciencias Ambientales de Aragón (IUCA), Universidad de Zaragoza, Zaragoza, Spain. [4]Alfred Wegener Institute, Helmholtz Centre for Polar and Marine Research, Bremerhaven, Germany. [5]School of Earth and Environmental Sciences, Cardiff University, Cardiff, UK. [6]Smithsonian Tropical Research Institute, Balboa Ancon, Panama. [7]Department of Earth Sciences, Natural History Museum, Cromwell Road, London, UK. [8]Carrera de Geología, Centro Universitario del Norte, Universidad de San Carlos de Guatemala, Cobán, Guatemala. [9]Dead Sea and Arava Sciences Center, Masada National Park, Mount Masada, Israel. [10]Ben-Gurion University of the Negev, Eilat Campus, Israel. [11]Ben-Gurion University of the Negev, Beer-Sheva, Israel. [12]S.J. Thugutt Geological Museum, Faculty of Geology, University of Warsaw, Warsaw, Poland. [13]Department of Earth Science, University of Bergen, Bergen, Norway. [14]Deceased: Martin A. Pearce. ✉e-mail: ndkrol@cyf-kr.edu.pl

ecological structure of marine environments, potentially driving changes on a planetary scale.

This study focuses on paleoceanographic changes in the Arctic Ocean (ArO), Atlantic, and Western Tethys (WT), driven by variations in marine connections around the ArO[2,8,9] and shoaling in the Central American Seaway (CAS), which separated the American continents during the Late Cretaceous[10–15]. Some evidence suggests that a semi-emergent island chain between the Pacific and Atlantic oceans may have existed about 70 Ma[12]. The tectonic evolution of the Caribbean Plate, which led to the formation of the oceanic plateau and volcanic arc, started even earlier and occurred over a longer period, contributing to the shoaling of the CAS well before the end of the Cretaceous. These long-term processes likely played a crucial role in the formation of a marine threshold, reducing the exchange of water masses between the Atlantic and Pacific oceans. While the exact timing of this transformation remains uncertain, geological proxies indicate a significant reorganization of ocean circulation patterns in the Maastrichtian. This is supported by the emergence of volcanic promontories on the Caribbean oceanic plateau[10,11], which helped create islands closely spaced between the Pacific and Atlantic oceans[12]. If the land connection was shallow enough to allow the exchange of terrestrial vertebrates between the Americas[13], the water exchange between the Atlantic and Pacific oceans must have been significantly restricted.

Interpreting the isolated palynological, micropaleontological, and geochemical data recovered from globally distributed sections, as illustrated in Fig. 1 and Supplementary Fig. 1(SF1), requires placing them within a broader global context, which can be achieved through numerical modeling. Therefore, we integrate the Earth System Model (ESM) simulations with geological proxies to enhance our understanding of the impact of paleogeographical changes on ocean circulation and climate transitions during the Maastrichtian. We examine contrasting scenarios, including CAS shoaling and variations in marine connections around the ArO, to identify the optimal fingerprint of global environmental shifts and compare their underlying effects with geological proxies. Building on this approach, we explore global patterns, providing valuable insights into how long-term disruptions in the salinity budget shape the chemical and physical properties of seawater, ultimately impacting marine primary producers that are particularly sensitive to such changes.

## Results

### Numerical model simulations

We utilize the fully coupled atmosphere-ocean ESM, COSMOS, to simulate key paleoceanographic scenarios for the Maastrichtian. These scenarios include shallowing in the CAS and restrictions in the marine connections between the ArO and the Global Ocean (GO). While these configurations have been previously proposed in the literature[2,8–14], their precise timing and potential global impacts remain poorly understood.

The shallowest water depth of the CAS is set to 10 m, ensuring that water masses flow through the CAS only from the Atlantic to the Pacific, preventing deeper Pacific waters from entering the Atlantic. This is essential, as it allows for the investigation of the effects of Arctic gateway opening and closure on salinity and temperature in the proto-North Atlantic (pNA), independent of Pacific water influence. Additional tested depths include 500 m, similar to the depth of the island arc in the western Pacific today. The deepest conditions tested range from 1700 to 2500 m, following reconstruction from the published literature[16]. Detailed information on the reasons for selecting these specific CAS depths can be found in the 'Methods' section. First, we examine changes in ocean current directions driven by various depths in the CAS and limitations in marine connections around the ArO. Our results show that water flow from the Atlantic to the Pacific oceans is restricted when the depth of the CAS is reduced to 10 m, regardless of constraints around the ArO (Fig. 2). Additionally, we investigate variations in global water salinity and temperature, both horizontally and vertically, driven by bathymetric changes in the CAS and around ArO (SF2). This experiment reveals that restrictions in the marine connections between the ArO and GO result in fresher surface waters outflow via the Greenland-Norwegian Seaway (GNS) (SF2 I). It is also evident that paleogeographical changes have a stronger effect on sea surface salinity than on sea surface temperature distribution (SF2 I), which is not the case for the deeper ocean, as shown for the depths 420 and

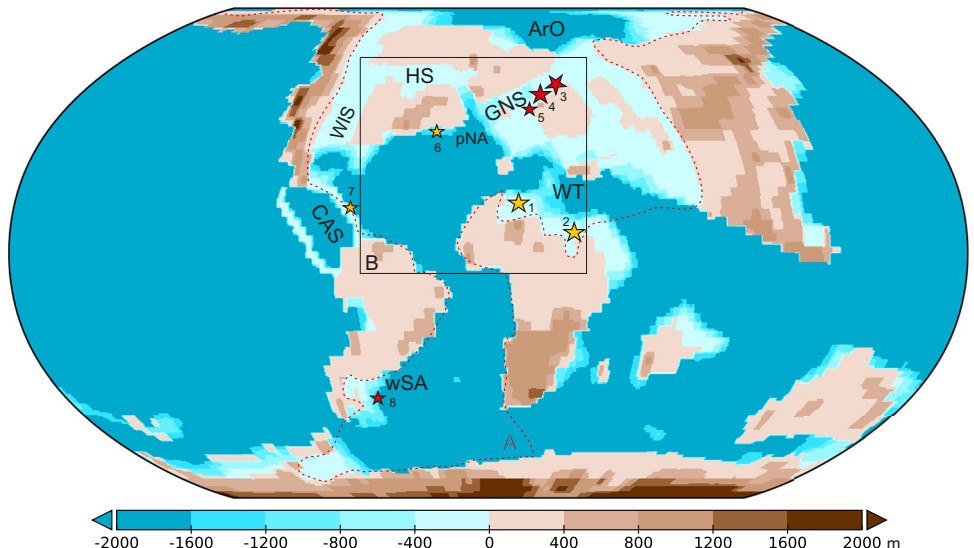

**Fig. 1 | Latest Cretaceous (Maastrichtian, 70 Ma) paleogeography[16] with a shallowed Central American Seaway (CAS) applied as a boundary condition in the Earth System Model simulations and interpolated to a resolution of 360 ×180°.** The geological sections analyzed in this study are indicated by large stars: (1) Sidi Ziane section in Algeria; (2) Re-2 core from the Negev region in Israel, both within the Western Tethys (WT); and two boreholes: (3) 6711/4-U-1 and (4) 6707/10-1, from the Greenland-Norwegian Seaway (GNS). Small stars represent additional published palynological data from locations such as (5) Barents Sea in the GNS; (6) Bass River south of the Hudson Seaway (HS); (7) Guatemala in the CAS; and (8) Bajada del Jagüel, Argentina, in the western South Atlantic (wSA). Remaining abbreviations: WIS – Western Interior Seaway, pNA - proto-North Atlantic, ArO - Arctic Ocean. The red dashed contour and black frame indicate **A** the area analyzed in the Earth system model simulations, encompassing the Western Tethys, Atlantic, and Arctic oceans, and **B** the analyzed region of the proto-North Atlantic.

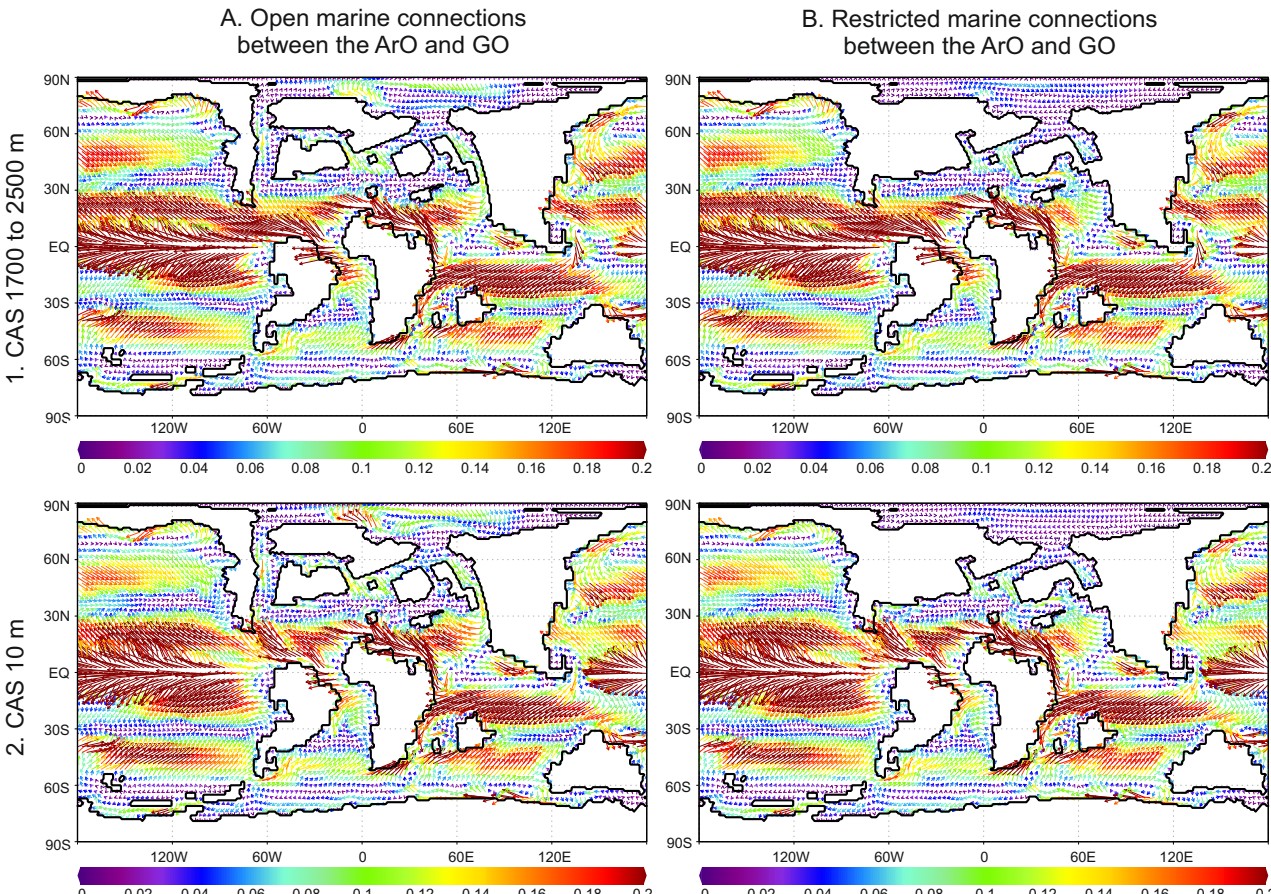

**Fig. 2 | Variations in global water currents driven by bathymetric changes in the Central American Seaway (CAS) and restrictions between the Arctic Ocean (ArO) and the Global Ocean (GO). A** Simulated ocean current patterns under open marine connections between the ArO and the GO; **B** Simulated ocean currents with restricted ArO–GO connection. Enhanced exchange between the Atlantic and Pacific oceans occurs under conditions of a deep CAS (1), whereas a shallow CAS (2) results in reduced inter-ocean exchange. The scale represents water current speed in meters per second (m/s).

960 m (SF2 II–III). Given that salinity variations can also result from shifts in the local precipitation-minus-evaporation (P-E) balance, we additionally simulate P-E in the North Atlantic–Arctic region under different gateway configurations between the Arctic and pNA oceans, as well as varying CAS depths (SF3). The results show that P-E differences between simulations are minimal and confined to localized areas. Therefore, it is highly unlikely that salinity changes in the North Atlantic region were driven by P-E variability.

To further explore the salinity trends, we analyze salinity and temperature transects within the ArO, pNA and WT, comparing various paleogeographic scenarios, including shallowing in the CAS and restrictions around the ArO (Fig. 3). Exclusive shallowing of the CAS increases sea-surface salinity in the ArO (Fig. 3.1A–C). Further limitation of marine connections around the ArO, restricting water-mass exchange between the ArO and GO to occur only through the GNS, yields a contrasting signal, resulting in decreasing sea-surface salinity in the ArO (Fig. 3.1D). Simulations with a CAS depth of 1700–2500 m show minimal differences in salinity in the Atlantic and WT (Fig. 3.2A), compared to simulations with a depth of 500 m (Fig. 3.2B). More pronounced changes in salinity stratification in the Atlantic occur when CAS depths are shallowest (Fig. 3.2C, D). Deep CAS and restricted ArO do not result in enhanced water column stratification. These findings suggest that the threshold for unidirectional flow from pNA to Pacific likely occurred between depths of 500 m and 10 m, within a paleodepth range supported by existing geological constraints (SN1).

Simulations of a shallow CAS with open marine connections around the ArO through the GNS, Western Interior Seaway (WIS), and

Hudson Seaway (HS), compared to a shallow CAS with restricted marine connections around the ArO, provide important additional insights about the ArO freshening (Fig. 4). In the shallow and restricted configuration, water-mass exchange occurs solely through the GNS. The decrease in sea-surface salinity in the ArO, likely driven by riverine run-off, results in an outflow of buoyant, low-salinity waters southward toward the pNA. These results support the finding that restricted connections between the ArO and GO, cause freshening of the ArO surface. We continue our investigation on the Arctic-Atlantic-Tethys salinity gradient by examining water transects within the GNS and the pNA, with a particular focus on both salinity and temperature changes (Fig. 5). Establishing the influence of CAS depth on ocean freshening and water column stratification is also essential.

In the scenario with open marine connections and a CAS depth of 10 m (Fig. 5.1A, B), water masses flowing southward from the GNS sink in the pNA, resulting in well-mixed conditions within the uppermost ~500 m of the water column. However, under ArO restrictions, mixing in the GNS is confined to a surface layer ~100 m thick (Fig. 5.2A, B), as freshwater masses from the ArO are too light to sink. Deepening of the CAS leads to significant changes in the salinity and temperature stratifications of the pNA (Fig. 5.3A, B). With a shallow CAS, water masses flow only from the Atlantic into the Pacific. Thus, mixing between surface and intermediate waters in the pNA depends solely on the salinity of water masses exiting the Arctic. Deepening the CAS allows an inflow of Pacific waters into the Atlantic in the southern part of the CAS, reaching the intermediate and deeper layers of the water column. These water masses are colder and less saline than the near-surface

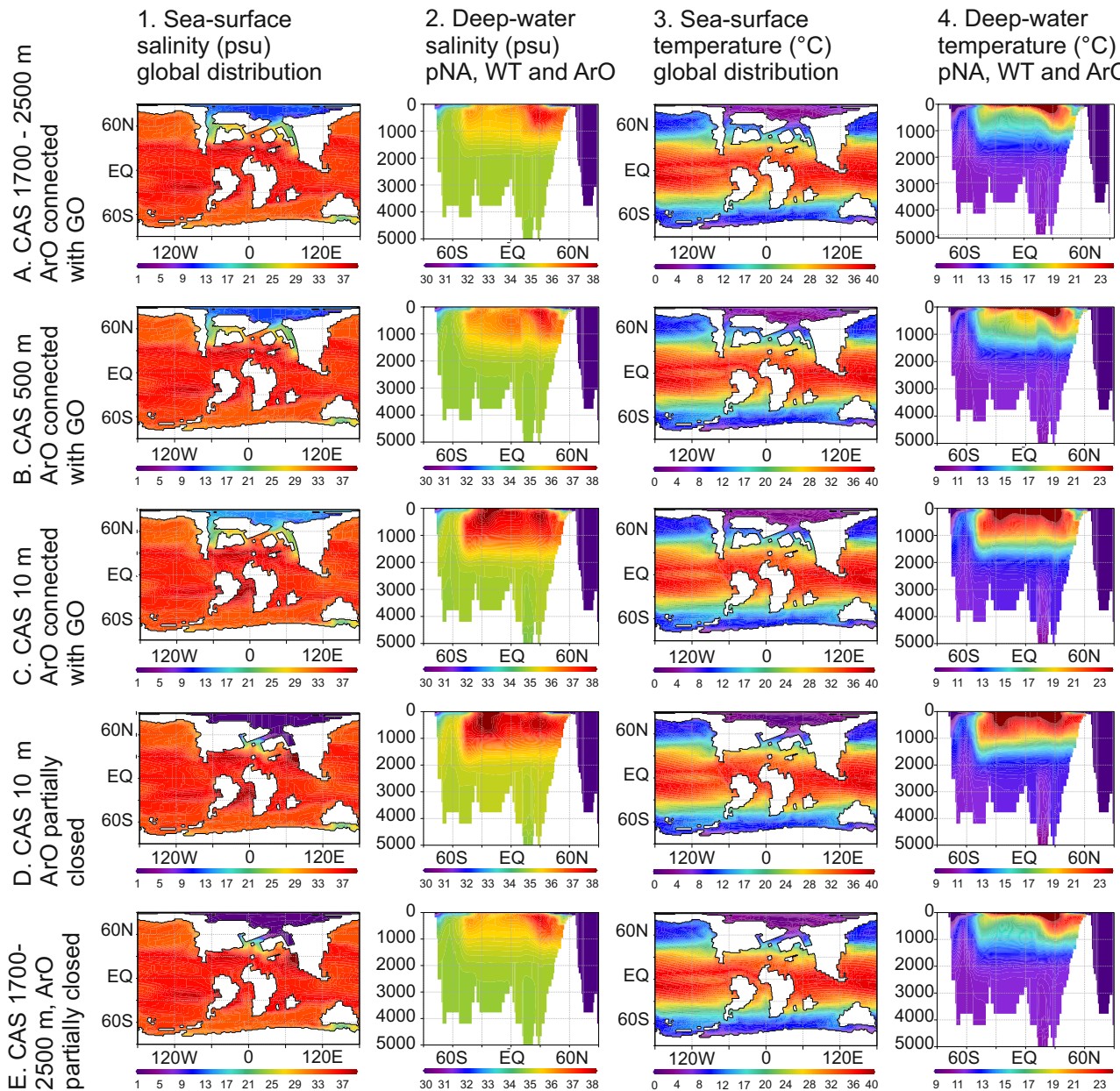

**Fig. 3 | A series of model experiments with varying paleogeographies and bathymetries.** The Arctic Ocean (ArO) is either restricted or connected to the Global Ocean (GO). Central American Seaway (CAS) depths are set to 10, 500, and 1700–2500 m. WT Western Tethys, pNA proto-North Atlantic. The analyzed region is shown in Fig. 1 (Region A).

waters, promoting the collapse of salinity stratification, particularly in the low latitudes (Fig. 5.3A). Therefore, only the combined effects of simultaneous Arctic gateway restrictions and CAS shoaling to depths that enforce unidirectional water flow from the pNA to the Pacific ensure increased water mass stratification in the pNA (Fig. 5.2). The 10 m CAS depth applied in our simulations serves as an experimental end-member, allowing us to isolate the effects of Arctic gateway restrictions on salinity and temperature in the pNA from Pacific water influence. While the specific CAS depth may vary, unidirectional flow–essential for pNA stratification in conjunction with Arctic restrictions–must be maintained.

## Paleontological and geochemical proxies under refined age models

The Sidi Ziane section in northern Algeria provides a continuous, and expanded sedimentary record of the uppermost Maastrichtian

(Figs. 6 and 7). This section offers an opportunity to examine changes in the biota and geochemistry that may represent responses to salinity variations, as investigated through our climate modeling. At Sidi Ziane, micropaleontological assemblages are mainly characterized by high relative abundances of the planktic foraminiferal genera *Heterohelix*, *Planohedbergella* and *Guembelitria*, as well as by a dominance of dinoflagellate cysts (dinocysts) from the family Areoligeraceae (Figs. 7–9). Within this section, we identify three significant foraminiferal events: 1) the last occurrence (LO) of *Gansserina gansseri* at 2750 cm below the Cretaceous/Paleogene boundary (KPB), marking the last 480 kyr of the Maastrichtian; 2) the first occurrence (FO) of *Plummerita hantkeninoides*, recorded at 1350 cm below the KPB, marking the last 100 kyr of the Maastrichtian; and 3) the LO of *Archaeoglobigerina cretacea* recorded at 100 cm below the KPB, marking the last 16 kyr of the Maastrichtian. Therefore, the Sidi Ziane section spans the last ~500 kyr of the Maastrichtian. This age estimate

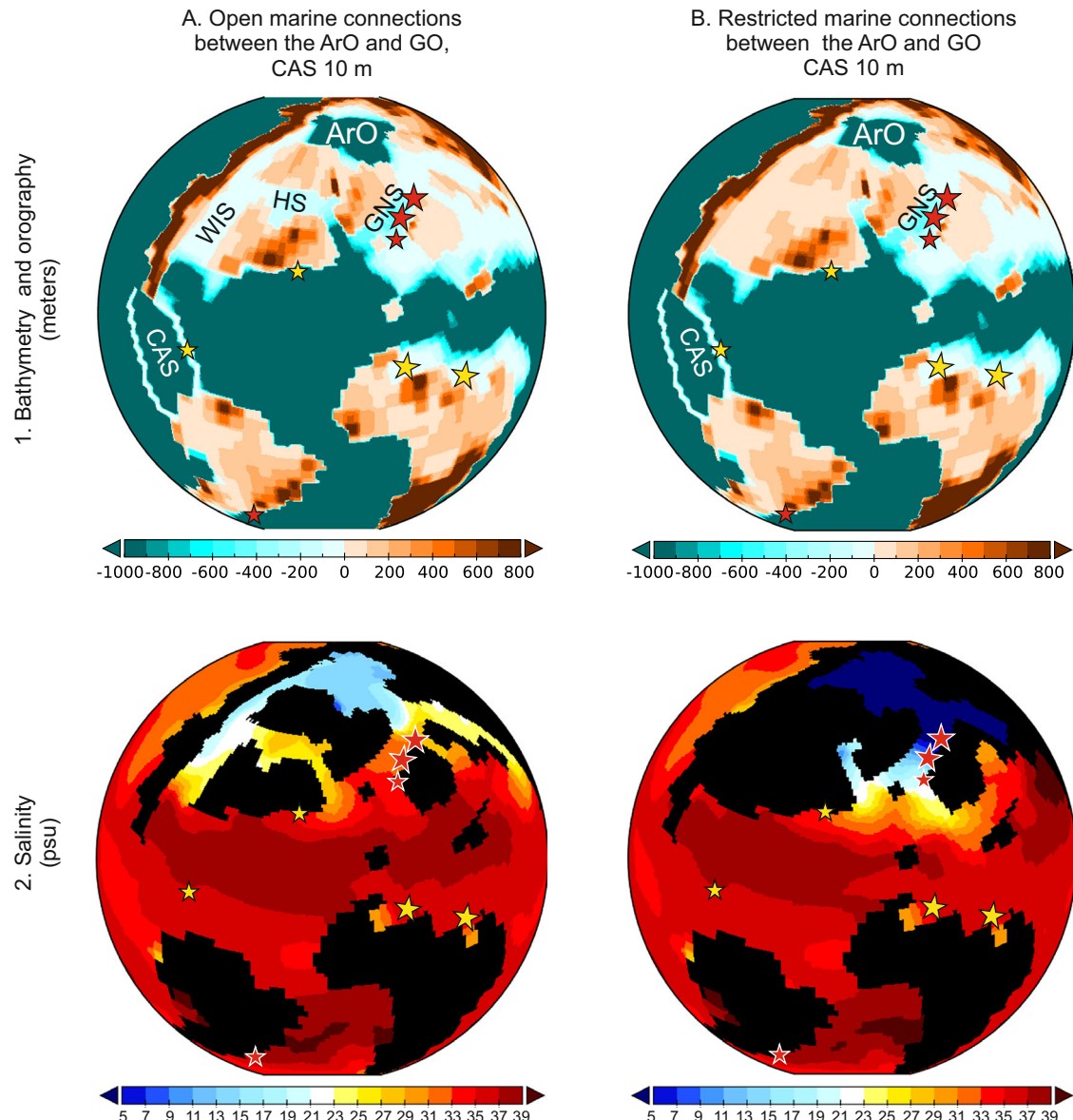

**Fig. 4 | The Maastrichtian Pacific-Arctic-Atlantic-Tethys salinity changes under different configurations. A** Shallow Central American Seaway (CAS) with open marine connections around the Arctic Ocean (ArO); **B** Shallow CAS with restricted marine connections around the ArO. WIS Western Interior Seaway, HS Hudson Seaway, GNS Greenland-Norwegian Seaway.

is supported by an acme of *Manumiella seelandica* (Fig. 7E), a dinocyst species typically occurring just below the KPB[17]. We refined this biochronological age model by establishing a chemostratigraphic correlation (Fig. 7) with the astronomically tuned section of Zumaia, Spain[18]. This approach enabled for (1) an independent validation of the age model and (2) an improved age resolution for Sidi Ziane section. Detailed explanations of the age model for Sidi Ziane and other studied localities are provided in the 'Methods' section.

For a more detailed comparison with the numerical model, bulk isotope stable analyses for complementary paleoclimatic and paleoenvironmental interpretations were carried out in the Sidi Ziane section. $\delta^{13}C$ values show three short-term carbon isotopic excursions (CIEs) superimposed on an overall positive trend, with CIE-1 occurring between -66.44 and 66.40 Ma, CIE-2 between 66.25 and 66.20 Ma, and CIE-3 between 66.17 and 66.11 Ma (Fig. 7A). The relationship of $\delta^{13}C_{bulk}$ with $\delta^{18}O_{bulk}$ shows a nonlinear correlation, $r = -0.159$, $\rho(\alpha) > 0.05$, suggesting that the primary trend is preserved[19]. It yields a clear trend showing progressive warming from ~66.5 Ma to 66.058 Ma, followed

by a noticeable cooling from ~66.051 to 66.021 Ma (Fig. 7B). The cooling is further supported by the acme of *M. seelandica* (66.047–66.040 Ma) (Fig. 7E), which is considered to have been particularly adapted to cooler surface waters[20].

The latest Maastrichtian Sidi Ziane section is dominated by areoligeracean dinocysts (Fig. 7E), similar to the mid-Maastrichtian section in the Negev region (SF1, SD1 and SD4). These cysts were produced by marine, single-celled phototrophic algae capable of adapting to low-nutrient conditions. A different palynological signal is observed at higher latitudes, in two cores from the Norwegian Sea: 6711/4-U-1 and 6707/10-1 (SF1, SD5 and SD6). The pNA localities are rich in peridiniacean dinocysts, which are produced by algae that thrive in nutrient-enriched environments, such as those influenced by upwelling or river runoff. These observations are consistent with the dinocyst record recognized at Bajada del Jagüel (Argentina, western South Atlantic[21]) (SF1, SD7). Overall, these geological proxies highlight notable differences among the studied low- and high-latitude sections and record enhanced fluctuations preceding the end of the Mesozoic.

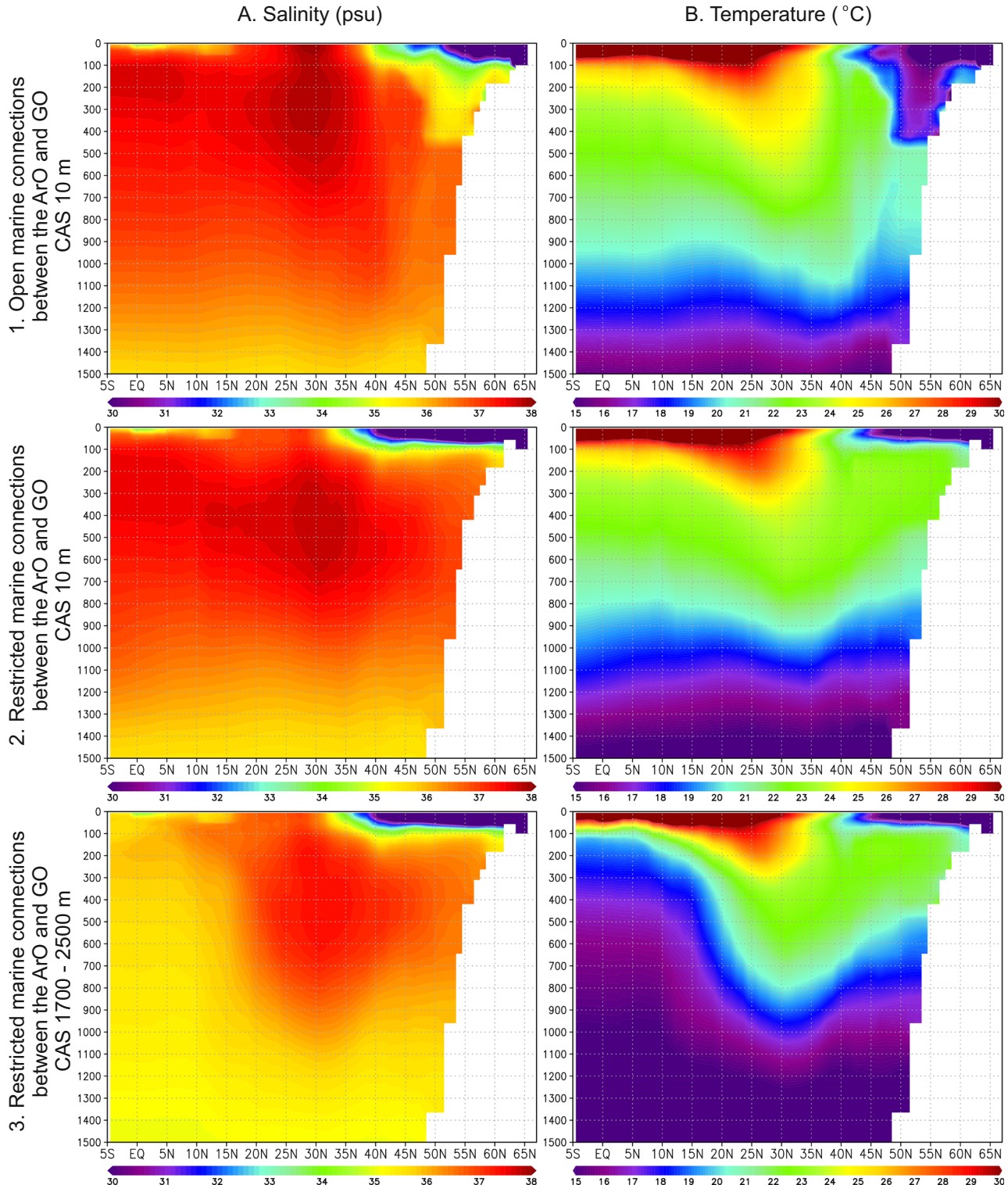

**Fig. 5 | Simulated salinity and temperature in the proto-North Atlantic under varying paleogeographic scenarios. A** Zonal mean salinity (psu); **B** Zonal mean temperature (°C). Various paleogeographic scenarios are considered, including both open and restricted connections between the Arctic Ocean (ArO) and the Global Ocean (GO), as well as variations in the depth of the Central American Seaway (CAS), ranging from shallow (10 m) to deep (1700–2500 m). The analyzed region corresponds to Region B in Fig. 1.

Additional information on foraminifera, isotopes and palynology is available in the Supplementary Datasets.

## Discussion

While the "Great American Biotic Interchange," resulting from the land bridge connecting the Americas, is thought to have occurred near the beginning of the Quaternary[22,23], geological evidence suggests that the obstruction of the inter-American region and the restriction of water mass exchange between the Atlantic and Pacific oceans occurred much earlier, in the latest Cretaceous[10,11]. Such changes could have caused significant shifts in water circulation patterns, affecting the marine ecosystem on a global scale. To explore this hypothesis, we analyzed

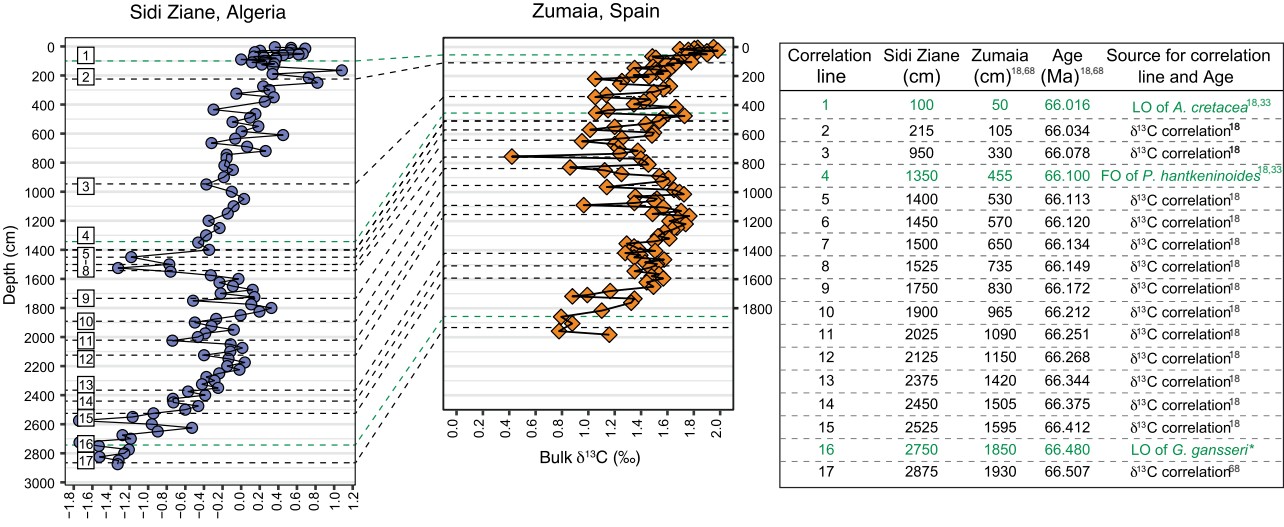

**Fig. 6 | Stratigraphic correlation between the Sidi Ziane and Zumaia sections, showing tie-points used for the age model of the late Maastrichtian at Sidi Ziane.** Last Occurrence Datum (LO) and First Occurrence (FO). *= Not

recognized at Zumaia, the LO of *Gansserina gansseri* is based on its stratigraphic position at Gubbio section, Italy[66].

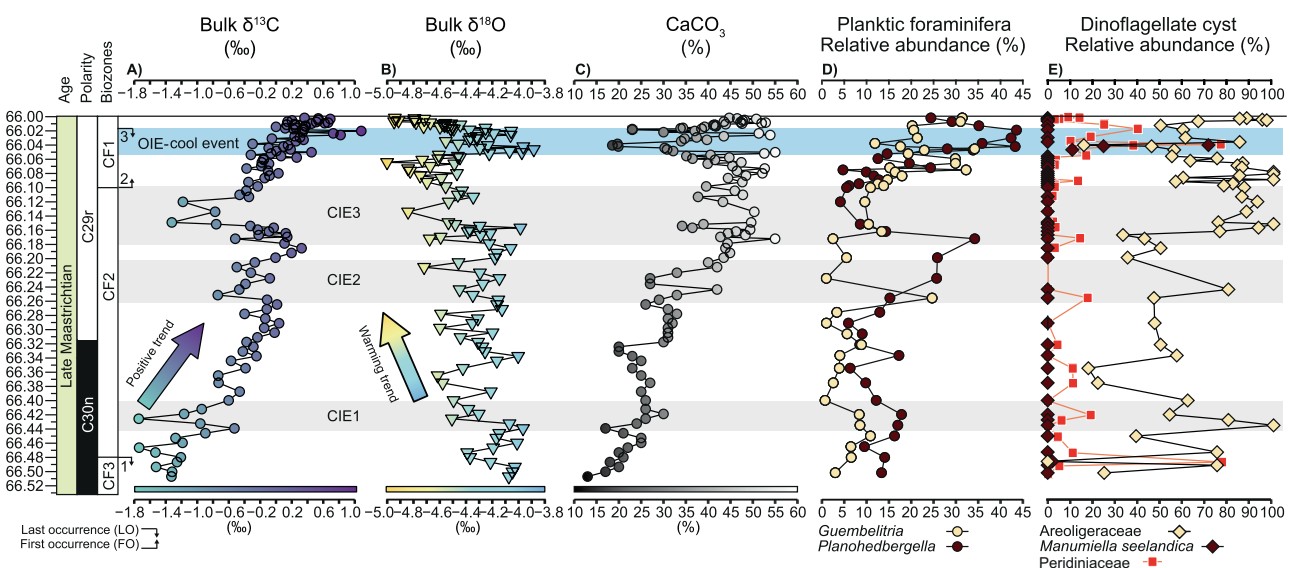

**Fig. 7 | Temporal correlation of planktic foraminifera, dinoflagellate cysts, and carbonate geochemistry of the Sidi Ziane section. A** Bulk sediment carbon isotope (δ13C, ‰); **B** oxygen stable isotope (δ18O, ‰); **C** calcium carbonate content (CaCO3 %); **D** Relative abundance (%) of selected planktic foraminifera genera;

**E** Relative abundance (%) of selected dinocysts. Biozone ranges are marked by: 1 = *Gansserina gansseri*; 2 = *Plummerita hantkeninoides*; 3 = *Archaeoglobigerina cretacea*. The age for the C29r/C30n reversal is based on the Geological Time Scale 2020[70].

geochemical and paleontological proxies, and conducted ESM simulations, drawing on literature-supported shoaling in the CAS[10–15] and progressive restrictions around the ArO[2,8,9].

## Modeled scenarios

Previous modeling studies of past environmental processes show that even small changes in the oceanic circulation pattern can have a strong environmental impact[24–26], especially within poorly oxygenated basins such as the Atlantic and Tethys[27]. Another study indicates better ventilated oceans triggered by changes in continental configurations, including (1) widening of the Atlantic Ocean, and (2) deepening of the Central Atlantic gateway[1]. So far, studies have not considered recent geological evidence for volcanically-induced shallowing of the inter-American area and restricted marine connections between the Pacific

and the Atlantic during the Maastrichtian, which constitutes a realistic scenario for the latest Cretaceous[10–15].

Our model simulations, which initially involve the shallowing of the CAS, result in restrictions on surface water masses exchange between the Atlantic and Pacific oceans (Fig. 2). In the case of additional restricted marine connections around the ArO, we observe a reduction in sea surface salinity (freshening) at high latitudes (Fig. 3.1D). While freshening in the ArO is primarily triggered by the restriction of marine connections to the GO, increased salinity stratification in the pNA and WT requires additional shallowing of the CAS (Fig. 3.2D). Salinity stratification also occurs with shallow CAS and open marine connections around the ArO; however, in this scenario, sea surface salinity increases (Fig. 3.1C), which contradicts our geological results. The reduction in throughflow in high-latitude

connections, driven by ArO restrictions combined with shallow CAS, leads to sea-surface freshening controlled by riverine run-off in the ArO, enhancing the southward outflow of low-salinity waters via the GNS (Fig. 4.2B).

More detailed insights into ocean freshening and water-column stratification are obtained through a comprehensive analysis of the zonal mean of salinity and temperature within the GNS and pNA (Fig. 5). These results confirm the significant impact of shallowing in the CAS and restricted marine connections around the ArO, which, when combined, lead to enhanced stratification within the GNS and pNA. We find out that salinity plays a key role in shaping the vertical temperature gradient. In scenarios with open marine pathways and moderately brackish sea-surface conditions in the GNS, the ocean is well-mixed, with no noticeable vertical gradient within the uppermost 500 meters of the water column (Fig. 5.1A, B). However, under ArO restrictions and higher fresh-water influx, mixing is confined to a shallower layer, approximately 100 m in thickness (Figs. 5.2A, B). This is because low-salinity waters become too light to sink, intensifying water mass stratification in the GNS, WT and pNA. The strongest deep water column mixing occurs in the case of the deepest CAS (Fig. 5.3A, B). These results clearly show that the most significant water-column stratification (the weakest mixing) occurs when restricted marine connections between the ArO and GO are combined with a shallow CAS (Fig. 5.2A, B). Neither CAS shallowing alone nor the closure of marine gateways between the ArO and GO leads to stratification as strong as when these factors are combined.

## Global dinoflagellate cyst signals from Maastrichtian sections

The available data from low-latitude locations, including the Sidi Ziane section in Algeria, the Re-2 core from Negev region in Israel, and Bass River in the USA, show the dominance of areoligeracean dinoflagellate cysts (SF1, SD1 and SD4, SD7). This group includes chorate, dorsoventrally compressed species of the *Glaphyrocysta*, *Areoligera* and other undifferentiated chorate areoligeraceans (Fig. 9), which appear in exceptionally high abundances in many low-latitude Maastrichtian sediments. A similar signal comes from the Maastrichtian of Guatemala, Central America, where these cysts, although not abundant, are represented exclusively by *Areoligera* and *Spiniferites* species[15].

Areoligeraceans are known to have been produced by phototrophic dinoflagellates inhabiting nearshore to offshore marine settings. However, offshore transport ('offworking') can also account for their occurrence in deeper environments. Many European sections dominated by areoligeracean taxa (and essentially absent of heterotrophic peridinioid taxa) are known to have been deposited in shallow, proximal sites, that typically experienced very limited continental runoff and were therefore, nutrient starved[28]. This group is often associated with marine, inner-neritic environments, preferring high-energy regimes[29,30]. The inner neritic environment typically refers to the shallower part of the continental shelf, extending from the shore to a depth of about 50 m[31].

Areoligeraceans have been reported as co-occurring with such low-salinity-tolerant dinoflagellate cysts as *Senegalinium*[32]. Similarly, in the El Kef section, Tunisia, these cysts are observed within the uppermost Maastrichtian and lowermost Danian, more commonly above the KPB, where they are superimposed on a higher abundance of freshwater algae[20]. This indicates a tolerance of this group for sea-surface salinity reduction. In addition, other authors associate all dorso-ventrally compressed gonyaulacoids (e.g., Areoligeraceae) with increased stratification[20,32]. Although the Sidi Ziane sediments may have been deposited at outer sublittoral to upper bathyal depths[33], the abundance of areoligeraceans likely reflects proximity to the paleo-coast and captures local signals of circulation reorganization preceding the end of the Mesozoic.

Different palynological assemblages are recorded in higher latitudes (cores 6711/4-U-1 and 6707/10-1, Norwegian Sea and Bajada del Jagüel, Argentina), where the Maastrichtian sediments are dominated by dinoflagellate cysts associated with the family Peridiniaceae (SF1, SD5 and SD6). This group is typically regarded as being produced by heterotrophic dinoflagellates, due to their morphological similarity to extant dinoflagellates of the heterotrophic family Protoperidiniaceae. It is likely that most peridiniaceans in the past were heterotrophs, and as a group, we consider them to represent a distinctive paleoenvironmental signal[34]. It has been shown that salinity has a primary influence on dinoflagellate cyst distribution[35]. Therefore, as with living protoperidiniaceans, they likely represent heterotrophic, nutrient-dependent organisms with a high tolerance for salinity fluctuations. Additionally, the Maastrichtian dinoflagellate cysts from high latitudes differ significantly from those representing the Campanian, being characterized by a richer and more diversified assemblage of 'normal' marine species[15,36,37]. This highlights the differing environmental conditions in the pNA near the end of the Cretaceous.

The overall conclusion drawn from the palynological record is that both high- and low-latitude signals, although expressed through distinct dinoflagellate cyst assemblages, indicate significant environmental shifts, including disruptions in ocean salinity budgets. The Maastrichtian sections at low latitudes are dominated by dinoflagellate cyst species characteristic of marginal or near-shore conditions, adapted to freshwater and reduced salinity environments. This suggests periodic, short-term reductions in shelf space and fluctuations in sea level. In contrast, the high-latitude dinoflagellate cyst record points to an increase in nutrient levels, likely derived from land and transported into the ArO via riverine fresh water runoff, which subsequently entered the GNS.

## High-resolution dinoflagellate cyst record from the Maastrichtian Sidi Ziane section

Data from the Sidi Ziane section indicate that, from the base to immediately below CIE1 (-1700 cm), cribroperidinioid cyst numbers are relatively high (SD1). This supports a proximal setting and suggests reduced salinity conditions, consistent with signals observed in globally dispersed sections. The genus *Spiniferites* appears to be well represented in this interval, but it is also present in the background assemblages above. *Spiniferites* represents a cosmopolitan shelfal group adapted to neritic and reduced salinity settings. The peridinoid *Palaeocystodinium* occurring towards the top and base of the section suggests a slight increase in nutrient levels, similar to the proto-peridinioid *Phelodinium* that represents in general a heterotrophic genus. The main taxa in this interval appear to be areoligeraceans and *Cribroperidinium* that suggest a shallowing trend along with a low salinity signal. Regarding the conspicuous spike in *Manumiella* around 325 cm (Fig. 7E), this species appears to be typical of relatively near-shore, inner-shelf marine environments, with morphological evolution of the cyst responding to changing water depths. Peaks in abundance of *Manumiella* (and closely related *Isabelidinium*) may indicate short-term regressions and/or ocean cooling prior to the KPB boundary. Similar 'spikes' in abundance of *Manumiella* associated with the KPB elsewhere in the world have also been related to mild cooling and regression[38]. This contributes to increasing evidence of significant marine regression during the Maastrichtian, possibly related to glacioeustasy during cooling climates after the mid-Cretaceous 'greenhouse'[39]. Such changes would enhance the isolation between the ArO and GO, increasing the influence of riverine freshwater input within the ArO.

## Salinity stratification revealed by palynology, micro-paleontology, and geochemistry

To evaluate the potential increase in water column salinity stratification in the WT, the palynological, foraminifera and geochemical records at Sidi Ziane were compared. A Principal Component Analysis (PCA) was conducted to robustly identify the most relevant planktic

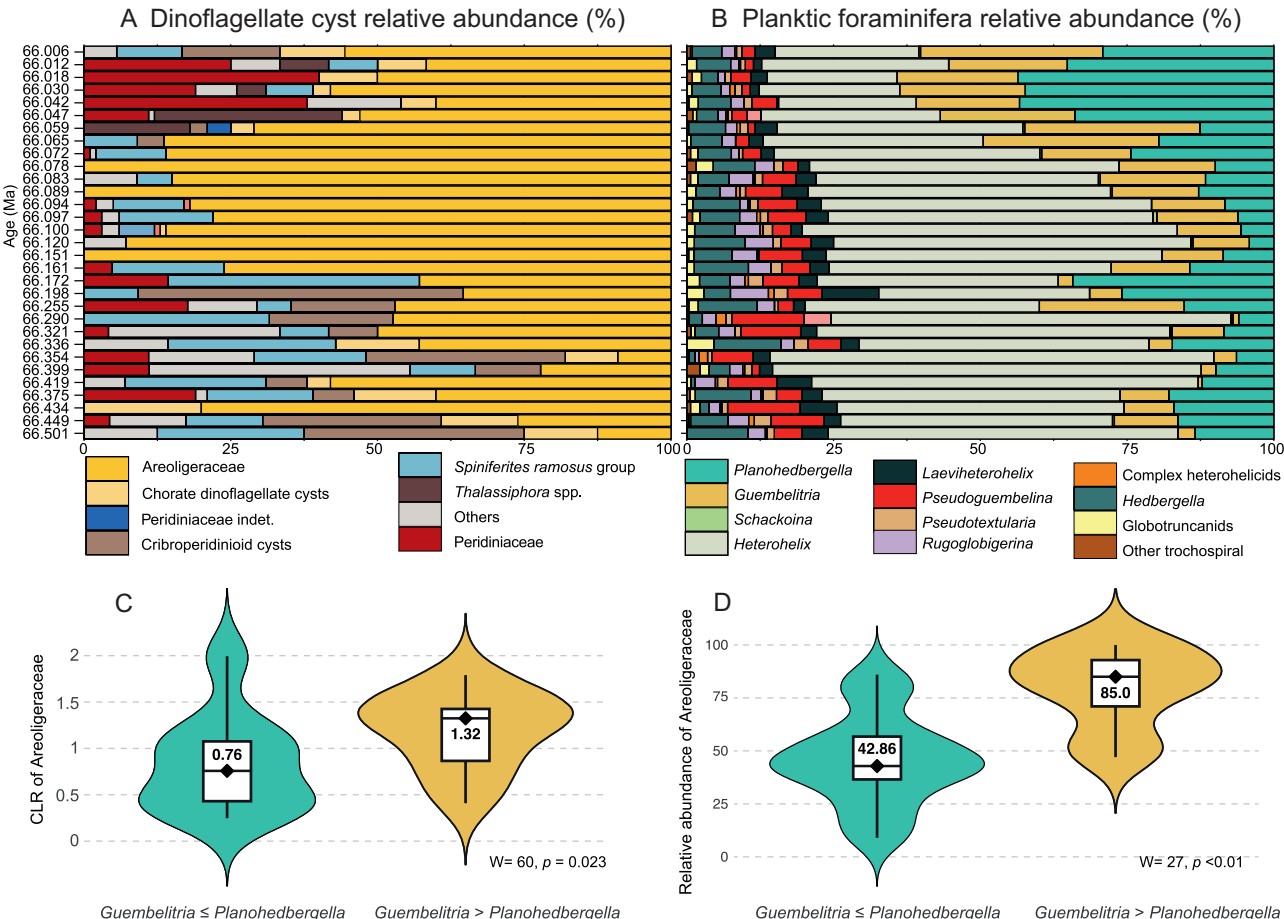

**Fig. 8 | Compositional data and statistical evaluation of the Sidi Ziane section, Algeria. A, B** Relative abundance of dinocyst and planktic foraminiferal groups in the directly comparable samples. **C** Violin plot for the compositional data analysis using CLR-transformed abundances, comparing the abundance of areoligeracean dinocysts under two different scenarios based on the relative abundance of the planktic foraminiferal genera *Guembelitria* and *Planohedbergella*. **D** Violin plot for the compositional data analysis using relative abundances, comparing the above mentioned groups. **C, D** also display the results of the Wilcoxon test to evaluate statistical significance.

foraminiferal genera for comparison with the dominant dinocyst group, the Areoligeraceae (Fig. 8A). PCA results revealed that PC1 explained 75.903% of the variance, while PC2 accounted for 17.02%, together explaining 93% of the total variance. The genera with the highest loadings in PC1 were *Heterohelix* (0.756), *Planohedbergella* (−0.534), and *Guembelitria* (−0.364). In PC2, *Guembelitria* had a loading of −0.791, while *Planohedbergella* had a loading of 0.594, suggesting potentially inverse ecological preferences. Loadings for *Heterohelix* were negligible in PC2. The remaining taxa exhibited minimal loadings in both PC1 and PC2. Since *Heterohelix* typically comprises more than 60% of the assemblages of the upper Maastrichtian[18,40], the focus was placed on the more specific ecological information provided by *Planohedbergella* and *Guembelitria*. A subsequent compositional data analysis (CoDA) was conducted to accurately compare these genera with the major dinocyst group, Areoligeraceae. Centered log-ratio (CLR) transformations were applied to the compositional data to prevent redundancy in relative abundance comparisons. The composition data analysis indicated that the Areoligeraceae exhibited significantly higher values when *Guembelitria* was more abundant than *Planohedbergella* (Fig. 8C). This relationship was statistically significant according to the Wilcoxon test, yielding a $p$ value <0.05 (0.02371). Furthermore, the same approach was applied using relative abundance data, which remained significant, with a $p$-value of less than 0.001 (Fig. 8D).

The approach outlined above, shows that the photo(auto)trophic Areoligeraceae yield a paleoecological signal similar to *Guembelitria*,

and somewhat inverse to *Planohedbergella*. High abundances of *Planohedbergella* correlates with higher δ[13]C and δ[18]O bulk values (Fig. 7A, B), and a positive correlation with higher $CaCO_3$ content (Fig. 7C), potentially representing higher productivity intervals. According to its isotopic signature, *Planohedbergella* is considered an intermediate-depth dwelling organism with seasonal preferences for thermocline depths and/or winter mixed layer depths[41]. This can explain why its higher abundances correlate to higher isotopic values. In contrast, the abundance in *Guembelitria*, which is considered to be a mixed-layer dweller tolerant to eutrophic, and dysoxic environments[41], is higher (>30%) when δ[18]O values are lowest (between -66.00 and 66.01 Ma, and between - 66.05 and 66.08 Ma). On this basis, we hypothesize that *Guembelitria* blooms at Sidi Ziane could be related to enhanced freshwater input from the continent, typically with low δ[18]O values[42], causing the stronger sea-surface stratification due to the contrasting densities of less and more saline seawater with a higher nutrient content, and a warmer mixed layer. However, the low δ[13]C values of this genus have also been interpreted either as evidence for a neustonic mode of life under poor nutrient conditions, or due to its own vital effects[43]. In any case, *Guembelitria* blooms are also often reported linked to blooms of other eutrophic taxa such as the calcareous dinoflagellate *Thoracosphaera/Cervisella*, or *Braarudosphaera*[44], and presented analyses of δ[13]C and δ[18]O on *Guembelitria*[43] reinforce its preferences for shallow and highly stratified waters. It appears that *Guembelitria* was an ecologic generalist and opportunist that tolerated stressed conditions with changes in temperature, salinity, oxygen and

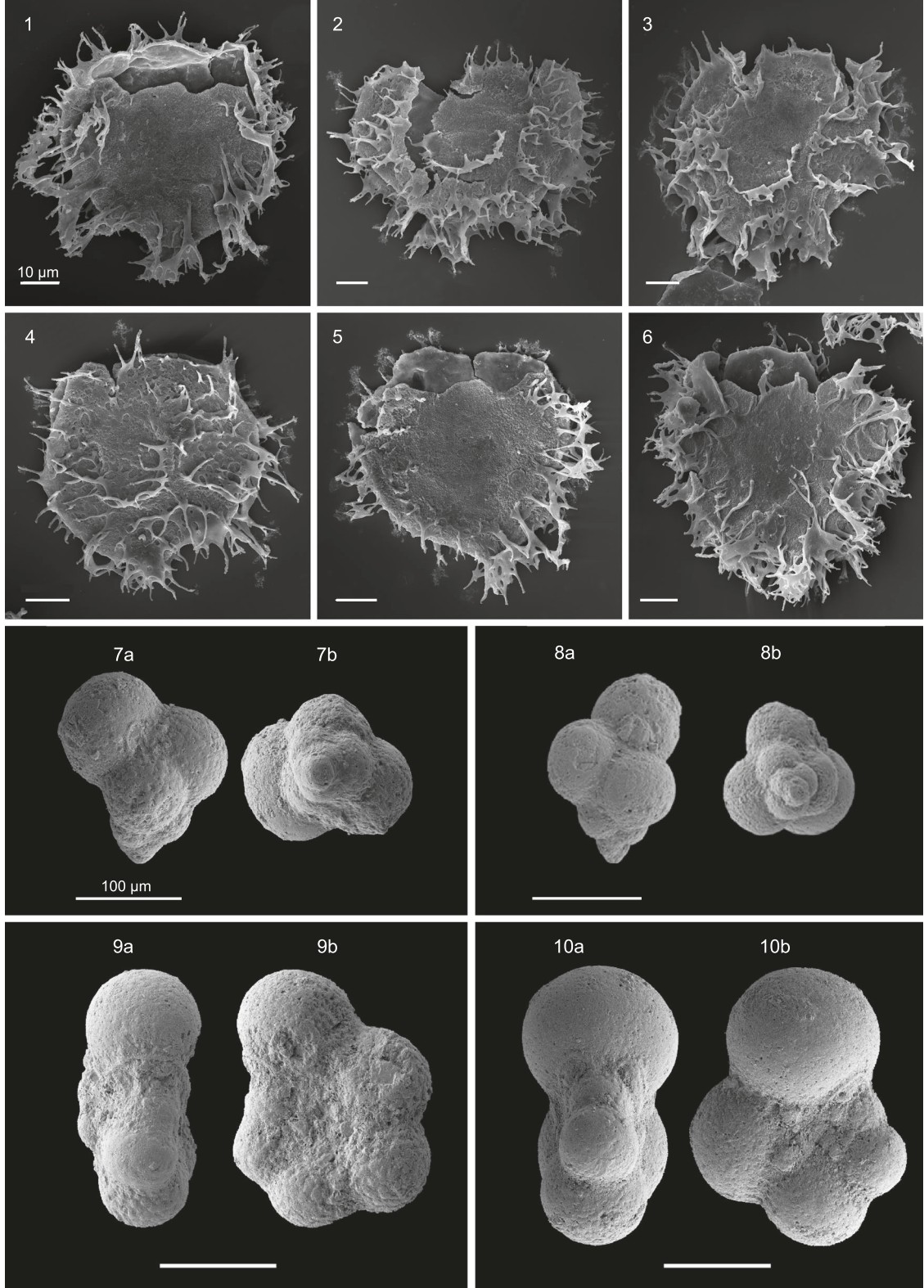

**Fig. 9 | Scanning Electron Microscope (SEM) photographs.** 1–6: Dinoflagellate cysts from Negev, Israel, belonging to the Areoligeraceae, which dominated lower-latitude regions during the latest Cretaceous. 1. *Areoligera volata*; 2–4. *A. senonensis*; 5–6: *A. medusettiformis*; 7–10: Planktic foraminifera from the uppermost Maastrichtian of Sidi Ziane, Algeria; 7–8: *Guembelitria cretacea*, a – axial view, b – spiral view; 9: *Planohedbergella prairiehillensis*, a – lateral view, b – umbilical view; 10: *Planohedbergella volutus* a – lateral view, b – umbilical view.

nutrients across the KPB[40,45]. In contrast, it should be emphasized that this species was rare under normal open marine conditions[40,46]. Accordingly, the Areoligeraceae dinocysts can be used as a potential proxy of high water column stratification in the context of more runoff and/or freshwater input. The palynological data from Sidi Ziane are consistent with the planktic foraminiferal, bulk $\delta^{13}C$ and $\delta^{18}O$ data (Fig. 7).

### Maastrichtian ocean freshening: evidence, missing pieces and implications

Can the freshening of the Atlantic and WT oceans be considered a feasible scenario for the latest Mesozoic world? Our numerical simulations, supported by geological Maastrichtian proxies and cross-referenced with published data, support this hypothesis. By integrating ESM results with palynological, micropaleontological, and geochemical evidence, we demonstrate that the shallowing in the CAS likely played a pivotal role in shaping global water mass circulation patterns. Additional paleogeographic changes that reduced throughflow in the high-latitude marine gateways contributed to sea-surface freshening in the ArO and the southward outflow of buoyant, low-salinity waters via the GNS. This, in turn, impeded the sinking of lighter water and intensified salinity-stratification processes (Figs. 3–5). Given the well-documented shoaling of the CAS and increasing restrictions on marine connections between the ArO and GO towards the end of the Cretaceous, we propose a scenario for the Maastrichtian characterized by a shallow CAS at a depth that enforces unidirectional westward flow, coupled with a restricted ArO. We argue that the combination of restricted marine gateways, which modified water mass exchange between the Pacific and Atlantic Oceans, and freshening in the ArO, played a defining role in the Maastrichtian. Our data indicate that these combined factors were sufficient to drive a major reorganization of Earth's climate system, with possible lasting implications for biodiversity and global environmental stability before the close of the Mesozoic.

Factors that could hinder our conclusions include the incomplete high-latitude geological record and the limited preservation of relevant latest Cretaceous paleontological material. These limitations arise from eustatic sea-level lowering and subaerial erosion of marginal and proximal sites along basin margins and platform areas[29,47–49], as well as restrictions around the ArO[2,8,9]. To address these gaps, we documented a unique, well-exposed, and continuous uppermost Maastrichtian sedimentological record from Algeria, in addition to other geographically dispersed Maastrichtian sections. Our findings indicate that low-latitude regions in the Maastrichtian, unlike the older intervals, were dominated by areoligeracean dinoflagellate cysts, suggesting marginal nearshore conditions and potentially increased water stratification. Peridiniacean cysts predominating during Maastrichtian in high latitudes point to enhanced nutrient availability, which suggests higher runoff, stronger terrigenous input and reduced-salinity environments. These findings support the notion that the co-evolution of large-scale hydrographic and ecosystem reorganizations in the pNA and WT, including paleo-seaway restrictions and modification of the ocean salinity budget, may have had a significant impact on Maastrichtian ocean dynamics.

While the lack of detailed temporal constraints on the end-Cretaceous paleogeography of the CAS, as well as the potential simultaneous openings of the Atlantic and Southern oceans during the Maastrichtian present challenges, these factors do not undermine the validity of our 'end-Mesozoic freshening' hypothesis. As shown by model simulations of near-future climate, the global system is highly sensitive to even minor changes in surface water density, which can disrupt thermohaline circulation and deep convection, leading to significant environmental shifts. While our results cannot be directly compared to modern changes, given that we simulate long-term processes, we observe that, in contrast to contemporary trends, pNA

freshening and the reorganization of ocean circulation do not need to rely on ice sheet interference. We demonstrate that such processes can be triggered independently by paleogeographic changes in ice-free conditions, as exemplified by the Late Cretaceous 'greenhouse' world.

The trajectory of ocean freshening during the Maastrichtian was influenced by two major events. Deccan Traps volcanism, which is estimated to have induced a 2 °C–4 °C increase in GO temperatures during the latest Maastrichtian[50,51], may have obscured a more distinct freshening signal. The Chicxulub impact, which triggered the KPB mass extinction[52] through GO acidification[53] and rapid cooling[20,54], not only caused widespread biotic turnover but also altered ecological niches, complicating the detection of long-term environmental trends such as ocean freshening during the early Danian.

Nevertheless, this study underscores the critical role of ocean gateway configurations and circulation patterns in shaping long-term ocean freshening, a phenomenon that remains insufficiently understood despite its global significance. The findings highlight the combined influence of CAS shoaling and progressive restriction of marine gateways around the ArO in driving substantial shifts in Earth's oceanic and climatic systems. Specifically, our results demonstrate that these changes led to (1) a large-scale reorganization of global water mass circulation, (2) a pronounced decrease in sea-surface salinity in the ArO and GNS and (3) salinity-driven water column stratification in the pNA and WT. Notably, the major restructuring of the water column in the pNA and WT occurs only when CAS shallowing reaches depths that establish unidirectional flow from the pNA to the Pacific, in conjunction with the progressive restriction of water exchange between the ArO and the GO. These large-scale modifications to the global salinity budget seem to have had significant consequences for marine surface biomass and productivity during the Maastrichtian, as indicated by the geological record. The 'ocean freshening near the end of the Mesozoic' scenario presented here fills a critical gap in our understanding of long-term GO freshening processes, showing strong agreement between numerical ESM simulations and global geological proxies.

## Methods
### Numerical model simulations
The ESM COSMOS used in the numerical simulations incorporates the ocean (MPI-OM) and atmosphere (ECHAM5) components. MPI-OM (Max Planck Institute Ocean Model) includes dynamic-thermodynamic sea ice model[55] and is run in the GR30/L40 configuration[56], with a formal horizontal resolution of $3.0° \times 1.8°$. ECHAM5 operates in the T31/L19 resolution[57] and has a horizontal resolution of ~3.75°. The high-resolution hydrological discharge model constitutes a part of the atmospheric component and simulates freshwater fluxes over the land surface[58,59].

The setup for the spin-up simulation (C-1120) has been previously described in detail[60]. In this simulation, the $CO_2$ level is set to 1120 ppm (4x pre-industrial (PI) levels of 280 ppm), while other greenhouse gases are maintained at PI levels. The solar constant is reduced by 1% relative to the PI level, and orbital parameters are set to 800 AD (beginning of a millennial run). No ice sheets are prescribed and the vegetation is fixed[61]. The simulation was run further to the year 14,800, with the depth of the CAS ranging from 1700 to 2500 m. Two sensitivity tests with different CAS depths and identical atmospheric land-sea masks as in C-1120, were performed. In the first experiment, the bathymetry around the CAS reflects recent constraints on the gateway depth, suggesting that the exchange of waters between the Atlantic and Pacific oceans was limited to a depth of 500 m. In the second experiment, the gateway depth was limited to 10 m, based on data indicating shallow connections between the Atlantic and Pacific oceans during the latest Cretaceous. All other boundary conditions, including depths of the marine gateways between the ArO and the GO were set identically to those in C-1120. Both simulations were branched off from C-1120 at year 10,800, and were run for 3000 years.

Additionally, we analyzed two more simulations with different atmospheric land-sea mask compared to C-1120. In both simulations, all gateways between the Arctic and the Atlantic oceans, except the GNS, were closed. The two simulations differ only in the CAS depth. In one, this depth is the same as in C-1120, while in the second, it is set to 10 m. All other boundary conditions remain identical to C-1120. The simulation with the deeper CAS (GNS-47_4x) was analyzed previously[60], and here, we extend the simulation by additional 800 years. The simulation with the 10 m CAS depth was restarted from the deep CAS run and extended for 3000 years. The analyzed data shown in Fig. 3 include the ArO, pNA and WT (Region A), while in Fig. 5 depicts the GNS (Region B), as outlined in Fig. 1.

## Simulated water depths of the CAS

The first assumption was that the depth of the Panama island arc could be comparable to that of an island arc in the western Pacific today (~500 m). This value seemed to be the most realistic estimate for the latest Cretaceous CAS. The selection of the deepest conditions (1700–2500 m) was supported by published paleogeographic data[16]. The shallowest depth was set to 10 m, ensuring that water masses flow through the CAS only from the Atlantic to the Pacific Ocean, preventing deeper Pacific waters from entering the Atlantic. This scenario assumed the presence of a shallow barrier, rather than fully emergent land, between the Atlantic and Pacific Oceans, enabling an assessment of its potential impact on the reorganization of oceanic circulation patterns. The estimated depth was inspired by the Maastrichtian dinoflagellate cyst record from Guatemala, CAS, which is characterized by areoligeracean morphotypes[15], suggesting deposition in an inner-neritic environment, typically interpreted as ranging from 0 to 50 m.

Even shallower depths are suggested by the widespread lateral extent of the Angostura Formation, which during the Maastrichtian formed a carbonate platform extending across Guatemala and Mexico[62]. For comparison, the present water depths of the Bahama carbonate platform generally do not exceed 25 m, with approximately 60% (61,400 km²) of the platform lying at depths of 5 m or less[63]. Similar conditions in the CAS seem plausible, as the Angostura Formation typically contains large benthic foraminifera (including alveolinids), which inhabit the seafloor of shallow, tropical, and subtropical seas in carbonate-rich environments, such as reefs and lagoons[62]. However, the aim of this study was not to estimate the exact depth of the barrier, but rather to observe trends in paleoenvironmental changes driven by shoaling in the CAS, identify the threshold triggering these changes, and compare the results with the geological record.

## Sections and samples studied

A total of 305 geological samples were analyzed for the purpose of this work. The last ~500 kyr of the Maastrichtian, preceding the Cretaceous/Paleogene boundary (KPB), were studied using 208 samples from the Sidi Ziane section in Algeria. These included marine palynology (54 samples), micropaleontology (41 samples), and geochemical signals (113 samples). Additional palynological analysis of 12 samples was conducted on the Maastrichtian Re-2 core from the Negev region, Israel. Cores 6711/4-U-1 and 6707/10-1, drilled in the Norwegian Sea, were re-analyzed from previous work[64], with interpretations based on 85 palynological slides. Additional data were obtained by reviewing the published record, which included various Maastrichtian sections globally, such as the Barents Sea[37] in the GNS, Bass River south of the HS[65], Guatemala in the CAS[15], and Bajada del Jagüel, Argentina, in the wSA[21]. The studied locations are illustrated in Fig. 1.

## Marine palynology

Cysts of marine single-celled algae (dinoflagellates) are composed of organic material that is highly resistant to taphonomic processes and thus constitutes a powerful tool to reconstruct past environments.

They were analyzed to identify differences between pre-Maastrichtian and Maastrichtian sediments, as well as to examine past oceanographic conditions in geographically dispersed sections in low- and high-latitude regions.

**Sample preparation.** Samples from the Sidi Ziane section, Algeria, and the Re-2 core, Israel, were processed, mounted, and analyzed at the Polish Academy of Sciences, Institute of Geological Sciences, Research Center in Kraków. ~20 grams of each sample were subjected to a rigorous cleaning process before being crushed into particles of about 2 mm in size. To eliminate carbonates and silicates, cold 38% hydrochloric (HCl) and 40% hydrofluoric acids were utilized. Heavy minerals were separated from light organic matter using a clock glass. Additional data from the Norwegian Sea cores 6711/4-U-1 and 6707/10-1[64], previously prepared using similar protocol, were re-analyzed for the purposes of this study.

**Morphological associations used.** The dinoflagellate cysts from the Algerian Sidi Ziane section were grouped into the following morphological categories: Chorate, dorso-ventrally compressed taxa including the *Glaphyrocysta-Areoligera* complex and other cysts with similar environmental preferences (SD1); Peridiniaceae group, including *Isabelidinium* spp. and specimens belonging to *Palaeocystodinium australinum-bulliforme* complex and *Manumiella seelandica*, both of which occur as influxes. Other cysts, such as *Phelodinium* spp., *Thalassiphora* spp., the *Spiniferites ramosus* group and cribroperidinioids, were treated separately. Additional groups were identified for the Re-2 core, including *Achomosphaera* spp., *Pervo/Exochosphaeridium* spp., *Thalassiphora* spp., and *Florentinia* spp.; and for the Norwegian Sea sections, including *Impagidinium* spp., and *Microdinium cretaceum*. For further details on the assemblage distributions in all studied profiles, see SF1, SD1, SD4, SD5, SD6, and SD7.

## Micropaleontology

Foraminifera are single-celled protists commonly found in variety of marine environments. Many of these organisms construct tests by secreting calcium carbonate. This micropaleontological group is abundant in the marine sedimentary record, making it a valuable tool for exploring past climatic and environmental changes. They were analyzed to establish a detailed age model for the Sidi Ziane section and to provide insights into past oceanographic conditions in the WT region.

**Sample preparation.** The samples were processed at the University of Zaragoza, Spain. They were treated with 10% diluted $H_2O_2$ for 2 h each. The disaggregated samples were then washed through a 63 µm sieve and dried in an oven at temperatures below 50 °C. Quantitative analysis was performed at the genus level, after obtaining a representative aliquot of ~300 specimens per sample using a microsplitter.

**Age model.** At Sidi Ziane, three planktic foraminiferal events have been identified: the LO of *Archaeoglobigerina cretacea* at 100 cm below the KPB, the FO of *Plummerita hantkeninoides* at 1350 cm[33], and the LO of *Gansserina gansseri* at 2750 cm (this paper). The most recent age calibration for the LO of *A. cretacea* and the FO of *P. hantkeninoides* bioevents are 16 and 99 kyr, respectively[18], and 490 kyr for the LO of *G. gansseri*[66], before the KPB. These bioevents provide a comprehensive temporal framework when integrated with the δ[13]C record, enabling detailed chemostratigraphic correlations. The δ[13]C-based correlation is a reliable approach, as demonstrated in several Maastrichtian sections across different latitudes and environments[67]. By correlating the δ[13]C curve from Sidi Ziane with the astronomically calibrated δ[13]C curve from the Spanish section of Zumaia[18,68], an age interpolation for each sample was performed, assuming a constant sedimentation rate between the anchored tie-points. The LO of *G. gansseri*, calibrated here

at 480 kyr, aligns closely with the previously assigned value[66], further supporting the validity of this age model. For detailed data on foraminiferal assemblages from Sidi Ziane, Algeria, see SD3.

## Calcium carbonate content
**Sample preparation.** The samples were analyzed at the University of Zaragoza, Spain. The calcium carbonate content of each rock sample was estimated using a manocalcimeter, which measured the carbon dioxide pressure rise produced by the acid digestion on the 113 rock samples. The analysis involved adding 5 ml of 5 M HCl to one gram of powdered sample in the reaction cell, and was conducted independently of the atmospheric pressure.

## Isotopes
**Sample preparation.** Measurements of $\delta^{13}C$ and $\delta^{18}O$ isotopes were performed on homogenized bulk powdered sediment from the 113 samples of the Sidi Ziane (SD2). The samples were analyzed at the Leibniz Laboratory for Radiometric Dating and Stable Isotope Research (Kiel University, Germany), using a Kiel IV carbonate preparation device connected to a MAT 253 mass spectrometer from ThermoScientific. During preparation, the carbonates were reacted with 100% phosphoric acid ($H_3PO_4$) under vacuum conditions at 75 °C, and the evolved carbon dioxide was analyzed eight times for each sample. All values are reported in Vienna Pee Dee Belemnite (VPDB) notation, compared to NBS19 and IAEA-603 standards. The precision of all different laboratory internal and international standards (NBS19 and IAEA-603) is <0.05‰ for $\delta^{13}C$ and <0.09‰ for $\delta^{18}O$ values.

## Multivariate and compositional data analyses
PCA was conducted to reduce the dimensionality of the identified planktic foraminiferal genera (variables) and to identify the main components that explain the variance in planktic foraminiferal assemblages across the entire studied interval. The PCA was performed using PAST software[69]. Following the PCA, CoDA was carried out to explore the relationships between the main foraminiferal components and the predominant areoligeracean dinoflagellate cyst group from directly comparable samples (i.e., those from the same stratigraphic horizon; Fig. 8). To overcome the constraint of the relative abundance compositional data (percentages), a CLR transformation was applied to the compositional dataset. The CLR transformation ensures that the taxa abundance data are appropriately scaled, allowing for more robust statistical comparisons. Both CoDA and CLR transformation were calculated using the specific package (compositions) in the RStudio software. Additionally, the non-parametric Wilcoxon test was employed to assess the statistical significance of the relationship between the main foraminiferal genera and the areoligeracean dinocyst group. This test was selected due to its suitability for comparing distributions without assuming normality, especially given the potential deviations from normality and some small sample sizes present in this study.

## Data availability
All information necessary to evaluate the conclusions of this study is provided in the main text and the Supplementary Files.

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

## Acknowledgements

This work was initiated as part of the internal PALEOCLIMATE project at the Polish Academy of Sciences, Institute of Geological Sciences, Research Center in Kraków. Geological data were partially processed using the Paleolog software developed by Michał Radmacher. VG, JAA, and IA were funded by grant PID2022-136233NB-I00, provided by MCIN/AEI/10.13039/501100011033 and the European Regional Development Fund (ERDF, "A way of making Europe"), as well as by the DGA Group E33_23R, supported by the Government of Aragon and the ERDF. VG also acknowledges funding from the Ministerio de Universidades (MIU) and the European Union through the Margarita Salas postdoctoral grant (funded by the European Union – NextGeneration EU), and the POSTUPV24/33 postdoctoral grant from the University of the Basque Country (UPV/EHU). DB gratefully acknowledges support from the National Geographic Society (grant no. GEFNE137-15). SAP thanks the Israeli Ministry of Science and Technology (MOST) for its ongoing support of the ADSSC. The authors acknowledge the use of the Servicio General de Apoyo a la Investigación (SAI), Universidad de Zaragoza. We thank Martin J. Head and Stijn De Schepper for their constructive comments on early versions of the manuscript, and Professor Stanisław Mazur, Director of the Institute of Geological Sciences, Polish Academy of Sciences, for his continued support and encouragement throughout the course of this work. The authors further wish to recognize the contributions of their colleague and friend, Martin A. Pearce, who passed away suddenly during the review process. His insight, dedication, and collaboration were integral to this study, and his memory is fondly honored in this publication.

## Author contributions

Development of the project idea: W.R., I.N., J.T., D.M.B., G.K., J.A.A. and I.A. Methodology: W.R., I.N., V.G., J.A.A. and M.N. Visualization: W.R., I.N., V.G., M.M., J.T. and M.N. Supervision: O.J.V., J.T., S.A.P., S.A., M.A.P. and G.M.

## Competing interests

We declare that the authors have no competing interests as defined by Nature Portfolio, or other interests that might be perceived to influence the results and/or discussion reported in this paper.
