## [Transparent Peer Review file · Nature Communications]

Ocean freshening near the end of the Mesozoic

Corresponding Author: Dr Wieslawa Radmacher

Version 0:

Reviewer comments:

Reviewer #1

(Remarks to the Author)

The article by W. Radmacher et al. investigates the freshening of the oceans in the late Mesozoic era, a crucial process in ocean dynamics with significant implications for marine biology. The study focuses on the period just before the K-Pg (Cretaceous–Paleogene) boundary. While previous research has primarily examined the effects of the Chicxulub impact and the Deccan Traps, the role of ancient seaways has largely been overlooked, despite the fact that their evolution likely had substantial impacts on ocean circulation and water mass distribution.

This manuscript seeks to address this gap by clarifying the influence of seaways on oceanic changes during this period. The authors employ climate modeling to analyze the seaways' effects and examine shifts in marine organism populations as indicators of environmental change. The use of these two approaches to support the argument is promising. However, some conclusions lack precision or are not sufficiently supported by the data. In particular, I find that the analysis of the climate simulations is somewhat superficial.

In its current form, this study cannot be published. It requires substantial revision, particularly in terms of clarity, precision, and support for its conclusions. The methodology itself is valid, but the Materials and Methods section lacks sufficient detail to allow for the reproduction of the digital experiments. The manuscript lacks precision and specific quantification in places. Additionally, some of the figures do not clearly illustrate the changes discussed in the text.

My comments:

The palaeogeographic framework used in this study is based on Warwick and Valdes (2004), where the Caribbean plate is bordered to the west by an island arc associated with subduction—a chain of islands separated by narrow seaways—and to the east by a second arc. The authors suggest that the shoaling of CAS may have restricted exchanges between the Pacific and Atlantic Oceans. According to studies by Burch et al. (2018), these low exchange rates potentially contributed to the Oceanic Anoxic Event 3 (OAE3) around 90–84 Ma. The authors propose that the shallow Central American Seaway (CAS), along with progressively restricted marine connections with the Arctic Ocean, limited water exchanges between the Pacific, Atlantic, and western Tethys.

The study focuses specifically on the late Maastrichtian period. However, the authors do not clearly specify whether these exchange restrictions, from a geological standpoint, were in effect during this particular time frame.

The initial results focus on depth sensitivity experiments for ocean passages, conducted using an Earth system model described in the Materials and Methods section. However, the boundary conditions, such as pCO₂, solar constant, and vegetation, are not specified. Based on Niezgodzki et al. (2019), we can infer that a pCO₂ concentration of 1120 ppm was likely used.

The authors test various configurations, including one with the Central American Seaway (CAS) at a depth of just 10 meters. The choice of the 10-meter Central American Seaway (CAS) scenario appears to align with the available data, but the authors do not explain whether such a scenario is realistic. Why was such a shallow depth chosen, and is this realistic given the presence of an island arc? More discussion is needed on this point to assess its plausibility. The second experiment combines a CAS depth of 10 meters with additional restrictions on Arctic Ocean exchanges. According to Niezgodzki et al. (2019), the Arctic passages were already quite shallow (<50 or 100 meters). However, the authors do not clarify the specific parameters they applied for this restricted Arctic configuration in their new experiment.

The authors note that experiments with a 10-meter-deep Central American Seaway (CAS) yield the most pronounced palaeoenvironmental changes, particularly in terms of water column stratification in the Arctic Ocean. However, it is unclear how these changes are illustrated in Figure 2C4. In general, Figure 2 is challenging to interpret due to the small size and limited contrast of the color scale, making it difficult to discern the changes described. The authors also mention that restricted exchanges through the Greenland-Norwegian Seaway increase temperature gradients in the Atlantic Ocean and western Tethys. However, this effect is hard to see in Figure 2D2, as the data appear to be globally averaged. The authors tested additional scenarios, with and without the Western Interior Seaway (WIS) and Hudson Seaway (HUS). They argue that the 10-meter CAS configuration best represents conditions at the end of the Cretaceous—specifically, the Maastrichtian.

On what basis is this depth chosen, and does this imply that these conditions were unique to the Maastrichtian? Additionally, what mechanisms could account for a change in CAS depth by the end of the Cretaceous?

In the second part of the study, the authors analyze multiple records, starting with the Sidi Ziâne section in Algeria and boreholes in the Norwegian Sea. However, some sections presented in the appendix (e.g., Argentina, USA) are never discussed in the manuscript.

Lines 211-214: The abundance of certain organisms is interpreted as indicative of shifts in temperature or nutrient-rich conditions, potentially driven by upwelling or runoff. The authors conclude that significant environmental changes occurred prior to the K-Pg boundary (KPB). Borehole 6707/10-1, which spans approximately 3 million years (~70–67 Ma), shows relatively minor variations (see Fig. S4B). In borehole 6711/4-U-1, there is a brief increase around 68 Ma; however, this occurs well before the KPB, and fluctuations between 68 Ma and the KPB do not appear markedly different from those preceding the 68 Ma event.

Lines 236-244: In the Discussion section, the authors suggest that a shallow Central American Seaway (CAS) (10 meters) increases water mass stratification in the Atlantic Ocean and western Tethys. However, there is no geological evidence supporting a continuous shallow barrier between North and South America; instead, geological data indicate the presence of an island arc associated with the subduction zone.

Lines 246-256: The authors propose that increased stratification occurred in the Atlantic Ocean and parts of the Tethys, yet they do not illustrate these changes in the figures. To support this claim, visual evidence of stratification changes should be provided. Additionally, the mechanisms underlying these changes—particularly the interactions between the CAS and Western Tethys—are not clearly presented or discussed.

Lines 261-266: The authors suggest comparing the scenario-induced changes (CAS uplift) with geological data and note that these processes coincide with global cooling and sea-level fall. However, the timeframe remains unclear. What specific period are the authors addressing? Based on palaeontological data, this study focuses on the Maastrichtian, a time marked by warming followed by a brief cooling just before the K-Pg boundary. The Earth has experienced global cooling since the Turonian, but the late Maastrichtian includes a notable temperature rise prior to this cooling episode.

Lines 291-343: The authors suggest salinity changes in the Sidi Ziâne section, possibly linked to increased continental runoff (L310). For the Norwegian Sea sections, they similarly conclude that runoff may have increased, contributing to water mass stratification (L341).

The authors propose that these salinity changes result from shifts in ocean circulation driven by evolving ocean passages. While the Norwegian Sea sections indicate a salinity response (although the figure is difficult to interpret), the authors do not clearly present salinity changes in the Tethys from their sensitivity experiments. Additionally, changes in salinity can also stem from shifts in the local precipitation-minus-evaporation (P-E) balance. Modifying ocean circulation could potentially alter atmospheric circulation as well—especially given the possible removal of the Western Interior Seaway (WIS) and Hudson Seaway (HUS)—which could impact the P-E balance. However, this point is not addressed.

The authors discuss the roles of the Deccan Traps and the Chicxulub impact, suggesting that these events may have masked the effects of ocean passage changes. The establishment of the Deccan Traps and the associated CO₂ emissions likely increased pCO₂ in the late Maastrichtian, leading to global warming, which could plausibly increase continental runoff. However, implicating the Chicxulub impact as a contributing factor seems more challenging, given that it occurred precisely at the K-Pg boundary, after the Maastrichtian.

While the authors emphasize the impact of ocean passage changes, there is insufficient geological evidence and temporal constraints to substantiate a causal link between these geological and oceanographic events. Why would the CAS have been shallow during or prior to the Maastrichtian? While some studies suggest a southern closure of the WIS, the northern portion of the WIS, along with much of the HUS, appears to remain open until the K-Pg boundary.

Reviewer #2

(Remarks to the Author)

Manuscript# NCOMMS-24-63460-T

Corresponding Author: Wiesława Marta Radmacher

Title: Freshening near the end of the Mesozoic?

Key Results

The key results from the paper, as I understand them, are that the model experiments show that shallowing gateways result in an enhanced freshening of the oceans in the Maastrichtian. This is similar to what has been found in studies of the Cenomanian and Early Eocene. This would suggest that such changes to the ocean system may be more common than currently apparent in the literature.

Validity

The climate model experiments show ocean freshening (lower salinity) with increased isolation of ocean basins. This is neither new nor a surprise.

Nonetheless, I believe this study to be a useful addition, but I am concerned by the following:

1. **Setup of the paper** – I re-read the manuscript many times and feel that the abstract and introduction would both benefit from being ‘tightened up’. By this I mean the following:

(a) **A more explicit statement of the aim of the study** – is the aim to show how palaeogeography affects the potential for freshening of the oceans in the geological past. That was my take.

(b) What then confused me as a reader were the statements about the importance of understanding climate change, the causes of extinction, palaeoecology, etc. Ok, yes, but these are not fully developed, and they feel like they have been added to the abstract and introduction to capture all the current buzz phrases. Yes, the potential for freshening of the oceans and its impact on ocean ecology and circulation is important. However, this study is about longer-term factors (viz. palaeogeography), whereas today, we are concerned with short-term drivers, like CO₂, land use changes and how these impact the hydrological system.

(c) **Extinction** – there is mention of the study interval preceding the mass extinction at the end of the Cretaceous (lines 33-35, 62-64). But a relationship between environmental changes (as described in this paper) and extinction is not then developed. I cannot see that they are related except for certain marine species being affected by salinity changes, if these are global, which does not seem likely. Is the intention to state that there are environmental changes leading up to the end Cretaceous which have not been examined enough?

(d) **Abstract** – this needs to be more formulaic: what I would do is to start by (sentence 1) stating that the study is aimed at understanding how common ocean freshening might be in the geological record (if that is indeed the aim); (sentence 2) How this might be important for future changes to the Earth ; (sentences 3 and 4) that the study does this by (1) running modelling experiments (long-term changes) and (2) outcrop sites to test model results in those areas; (5) That the results indicate that freshening may have occurred in the Maastrichtian.

2. **Modelling experiments** - The validity of using a 10m bathymetric experiment for the Central American Seaway and using this to make conclusions. Regarding climate modelling, a 10m water depth is essentially land! It is also unlikely to be the case for the entire length of the arcs that form the barriers. The authors could do no worse than look at the present-day bathymetries of the island and continental arcs in the western Pacific. Especially the discontinuous nature of some of these. Do I think that the Central American Seaway has barriers to deep water flow based on the geology I know? Certainly! I, therefore, agree with the authors. But, I do not consider this new, although it is useful.

3. **The outcrop data** – this section is fine and, to me, is a paper in its own right, especially the Algerian site. However, for the general aim of the paper, if I understand this correctly, I am concerned that all the localities used in the study to indicate freshening of the oceans are near continental areas and are therefore much more susceptible to changes in freshwater fluxes. They need not indicate global changes (though I think this is what the authors hope through the isotope data? If so, this needs to be drawn out more clearly).

4. **The equation of ocean freshening and stratification** (e.g. line 343). Although freshwater fluxes to the ocean can and do lead to water column stratification, in other circumstances, they can result in instability and over-turning.

5. **“...the effect of salinity is often speculated or even overlooked”** (line 334). Is this really true? There are numerous papers on density stratification and the role of salinity on circulation, going back to at least 1982 and Brass and 1998 and Hay et al. More recently, this has been discussed by Ladant et al. (2020), which the authors include in the reference list.

Significance

The conclusion that palaeogeography may be important for restricting ocean basins and enhancing the potential for ‘freshening of the ocean in these basins’ is not new, but it is useful. The important role of gateways on palaeoclimate has long been known (see Berggren 1982 as a start). That the authors identify this for the Maastrichtian to add to previously published work on the Cenomanian and Early Eocene is, to me, a useful addition.

However, I am less clear on how the outcrop data fits into testing this, given my concern (see above) that the localities are all near-shore and susceptible to local hydrological changes. This might be an area where the authors could strengthen the arguments using the same data.

Data and Methodology

Including observational data and model experiments is a common and powerful approach for examining the response of the

Earth system to changes in boundary conditions. To that extent, the data and methodologies are valid. However, I am struggling with the detailed analysis of the Algeria site data and the bigger-picture question being discussed. This may be me, but there feels like a disjunct here. It is possible that this could be addressed by rewriting the section or turning the whole paper around, viz., “how the analysis of the Algeria data suggests changes to the ocean system and hydrology, that are then examined using the modelling”, rather than the modelling followed by the data. This is just a thought.

I am less experienced with biostratigraphy and palynology, but from what I do understand, this all seems valid.

The modelling relies on one published Maastrichtian palaeogeography. What was the rationale for using this one? There is an updated version to this map if the authors are interested.

Analytical approach

The analysis of the biostratigraphy data is outside of my principal area of expertise.

Suggested improvements

My recommendation from reading this manuscript is that it would greatly benefit from a more explicit statement of the project aims and objectives and a clearer link between the aims of the Algeria analysis and the big-picture modelling study.

Clarity and context

1. Although the editorial guidelines say not to focus on grammar, I would urge the authors to avoid using words and statements such as the following:

(a) “*favouring*” (line 35) – I would be more direct, for example “indicate”, “show”.

(b) “*Seem to*” (line 39) – instead use “...processes may have played...”

(c) “*We assume that*” (line 152) – either your data shows this, or your assessment of the existing literature shows this.

(d) “*Reliable*” (line 154) – what do you mean by “reliable”? Is this the view supported by data, or the most recent publication, or something else?

(e) “*Novel*” (line 230; but used several times in the manuscript) – In this case, this is not a novel (meaning “new”) mechanism. “Novel” should only be used if the mechanism has not been recognized previously.

2. **Title:** I would add the word “Ocean” at the beginning of the title to make the context of “freshening” clearer to readers.

3. When you use “*Open-marine*” (Line 113), do you mean deep-water connectivity? The system is open to the global ocean system, just at shallow depths. For example, the Atlantic and Pacific were not separated in the Late Cretaceous as they are today by Central America.

References

There is a comprehensive reference list, although there are a few papers I think the authors should have a look at, for example, Laugié et al. (2021), <https://agupubs.onlinelibrary.wiley.com/doi/full/10.1029/2020PA004202>

Overview

This manuscript describes some very interesting results, and there is much that I think would be useful to see published. However, as I have described above, I think the paper needs some reorganization and further thought. I encourage the authors to make changes, and I wish them luck in their research.

Reviewer #3

(Remarks to the Author)

This study integrates modelling results with micropalaeontological and geochemical data from four geological sections in a time slice near the end of the Mesozoic (around 66Ma), just preceding the K-T extinction event. Using the COSMOS Earth System Model, Radmacher and co-authors test five scenarios with varying depths of the Central American Seaway (CAS) and marine connectivity around the Arctic. By shoaling the CAS, the model shows an increased salinity stratification in the proto-Atlantic, an effect that is exacerbated when restricting seaway connections around the Arctic. The authors compare the modelling results with previously published micropalaeontological and geochemical data (supplemented with additional analyses) from four localities: the Sidi Ziane Section (Algeria), the Re-2 Core from the Negev region (Israel) and two Norwegian Sea cores. They report increased relative abundances of dinoflagellate cysts belonging to the Family Areoligeraceae which is surmised to be a result of increased stratification. Increased abundances of protoperidiniaceans, known heterotrophs, are interpreted as evidence for the presence of nutrient-rich water prone to salinity fluctuations. Taken together, the authors conclude that their modelling and data point to a freshening at high latitudes with an increased outflow of low salinity waters through the Greenland-Norwegian Seaway.

The authors should be commended because they present results from a united effort of modelers and data driven scientists in a single study which will therefore be of interest to the diverse readership of Nature Communications. I find the manuscript easy to follow, but found a number of grammatical and formatting issues listed as comments in the annotated pdf (attached).

The text is accompanied by clear figures that effectively convey the authors' message.

Major Comment

Relative abundances are a form of compositional data (their total summing to 1) meaning that a rise in one species is automatically at the expense of another (the constant-sum constraint). In figure 6, selected profiles of planktic foraminifera and dinoflagellate cysts are presented which are difficult to interpret without knowledge of the whole assemblage composition (which the authors nicely present in the figures of the supplementary data). This leads to interpretations that are not necessarily backed up by the data. For example, on Line 294–295 the authors claim “when *Guembelitra* is more abundant than *Planohedbergella*, *Areoligeraceae* show higher abundances”. This is clearly not the case in CIE1 where areoligeraceans are close to 100% of the palynological assemblages while relative abundances of *Planohedbergella* exceed those of *Guembelitra*. Such covariance as suggested as suggested by the authors on lines 294–295 can only be examined using rigorous statistics that obey the laws of compositional data analysis.

In addition, relying on changes in relative abundance, especially for dinoflagellate cysts, without knowledge of the burial flux or concentration of cysts can be challenging. Dinoflagellate cyst burial flux calculations are recommended (they can be calculated if the marker grain method has been applied during preparation) as they can separate rises of number of cysts that are result of increased incorporation of cysts in the sediment from those that are the result of changes in the sedimentation rate. If these data are available, they would be invaluable in interpreting the presented assemblage changes.

Other comments

Abstract: why is there no mention of any of the geochemical and micropalaeontological data in the abstract? The manuscript discusses an integration of modelling and data but the abstract only mentions modelling results. I believe this is a particular strength of the manuscript and should be included here.

Line 54: Reference 21 deals with the late Neogene and offers no evidence about the latest Cretaceous as the authors seem to suggest here.

Line 196 and Figure 6A: CIE1 (high $\delta^{13}\text{C}$ corresponding to low $\delta^{18}\text{O}$ and peak in *Areoligeraceae*) behaves differently than the other CIEs where lower than expected $\delta^{13}\text{C}$ are recorded. This is not explained in the text. Later in the text these CIEs are ignored even though they were deemed important enough to highlight here and in Figure 5.

Line 268: what about preservation of calcium carbonate in cooler waters? This seems like it could be an issue near the bottom of the Sidi Zaine section where % CaCO_3 is very low. Will this not impact foraminifer preservation and the associated stable isotope signal?

Line 287: “this group can be considered as indicative of stressed near shore marine environment, with notably fresh-water input and changes in surface-water salinities followed by enhanced water stratification.” This seems like an unrealistically specific condition for increased abundance of areoligeraceans. I agree the data shows raised abundances could be indicative of “stressed” near shore marine environments but “notably fresh-water input and changes in surface-water salinities followed by enhanced water stratification” seems like an overreach. It may be the palaeoenvironment suggested in the current study, but I think this must be rephrased as not to sound that an increase in areoligeraceans is automatically indicative of only this specific situation.

Lines 292–297: I find this very speculative, and this is definitively not the case for CIE1 were abundances of *Planohedbergella* are higher than *Guembelitra* in the same sample when areoligeraceans form close to 100% of the dinoflagellate cyst assemblage (Figure 6 D–E). An inference like the one made by the authors in the sentence starting at line 293 (*Areoligeraceae* showing an inverse paleoecological signal with respect to *Planohedbergella*...), should be backed up by multivariate statistics.

Line 302–303: higher productivity could indeed result in higher CaCO_3 but post-deposition dissolution is an important factor in this, especially in cooler high latitude water (e.g. Henrich et al., 2002). Again, calculating a dinoflagellate cyst burial flux would give a much more reliable estimate of the organic matter burial rate and therefore productivity.

Line 309: “*Guembelitra* bloom”. A high relative abundance is not evidence of a bloom which suggests a high burial flux (high productivity).

Line 327–328: Reference needs replacing with reference number (Harris and Tocher, 2023).

Line 336–337: sentence needs rephrasing.

Line 359: I agree with the authors here. Only relative palaeo-salinity changes can be derived since the exact composition of the Maastrichtian seawater is not known. These relative palaeo-salinities can be derived from planktonic foraminifer oxygen isotope records corrected for global ice volume using a normalised global compilation of benthic oxygen isotope records. This has been done for the Pliocene–Quaternary but I am not aware of an equivalent for the Maastrichtian. If available, such a calculation could help solidify the inferred freshening. As the authors point out (e.g., line 396), the studied time interval is ice-free and a conversion of the planktonic oxygen isotope signal using the Shackleton (1974) equation could yield a record that is mainly influenced by salinity changes. Maybe this can be used to give a rough idea of salinity changes?

Figure 2: check the layering of the red rectangle. Some of the differences discussed in the text are hard to discern at the scale of the individual maps. Add dimensions to the colour gradient variables along the X axis.

Figure 2: “Caribbean Seaway” at the top of the figure is the same as “Central American Seaway” in the caption and text?

Figure 3: add dimensions to the colour gradient variable displayed along the bottom of each figure. HS = Hudson Seaway (correct?). Note that in line 114 the Hudson Seaway was abbreviated HUS instead of HS. Make uniform.

Figure 3A corresponds to the same model parameters employed in Figure 2A (Deep CAS with open Arctic marine connections) while Figure 3B corresponds to Figure 2D (Shallow CAS with reduced Arctic marine connections).

Figure 4A shows the difference between Open GNS/shallow CAS Vs. Open GNS, WIS and HUS/shallow CAS. Figure 4B shows the difference between Open GNS/shallow CAS Vs. Open GNS, WIS and HUS/Deep CAS.

References

Henrich, R., et al., 2002. Carbonate preservation records of the past 3 Myr in the Norwegian–Greenland Sea and the northern North Atlantic: implications for the history of NADW production. *Deep-Sea Research Part I-Oceanographic Research Papers* 184, 17-39 10.1016/S0025-3227(01)00279-1.

Shackleton, N.J., 1974. Attainment of isotopic equilibrium between ocean water and the benthonic foraminifera genus *Uvigerina*: isotopic changes in the ocean during the last glacial. *Colloques internationaux du C.N.R.S.* 219, 203-210

Version 1:

Reviewer comments:

Reviewer #2

(Remarks to the Author)
Reviewer Assessment

Overview

Having reviewed an earlier version of this manuscript, I must thank the authors for the changes they have made. These changes have addressed many of the concerns I raised. Consequently, my review will be short and focus on any remaining or new points.

The title, abstract, and set-up are much stronger, and the balance and link between the modelling and observational data are stronger. I was particularly interested in the modelling result showing the need for changes in both the Arctic (North Atlantic) and Caribbean gateways to generate significant changes in salinity, which was much more clearly stated in this version.

Noteworthy results

1. The modelling suggests that to maximise the freshening of the Central Atlantic—western Tethys ocean, circulation changes at the Arctic and Caribbean gateways need to be limited. Looking at the impact of multiple tectonic and palaeogeographic changes is important and, in my view, has long been overlooked. Too often, the focus has been on investigating a single factor, so this is good to see here.
2. In the modelling, changing the sill depth of the Caribbean gateway only has local effects on precipitation-evaporation, P-E (lines 135-136). I find this interesting, because as the authors note, it is changes in freshwater runoff that are important in the Arctic for salinity changes there. In contrast, regional changes in P-E have been shown to result from modelled changes in atmospheric chemistry or orbital changes.
3. There is a change in the planktonic assemblages during the Maastrichtian in the low-latitude sites, including the new site in Algeria that the authors discuss at length (lines 480-481). This may indicate increased stratification.

Significance

The climate model experiments show ocean freshening (lower salinity) with increased isolation of ocean basins. As I said in my original review, this is neither new nor surprising. However, the authors have emphasised in this version the importance of changing both the Arctic and Caribbean gateways to generate freshening. This is interesting and certainly needs to be published.

Does the work support the conclusions and claims, or is additional evidence needed

The observational data analysed and described in this paper are consistent with the hypotheses drawn from the modelling experiments.

Whether the data provide unequivocal evidence supporting the posed hypotheses is another matter. However, the only way anyone could address this problem would be to generate planktonic biome maps for the Atlantic using the deep-sea drilling data. However, I concede that correlating these data would not be easy with the fine temporal resolution required.

Are there any flaws in the data analysis, interpretation and conclusions? Do these prohibit publication or require revision?

My only reservation is the degree to which the salinity changes in the biotic signal reflect local rather than regional salinity. However, in response to this comment in my original review, the authors pointed out the importance of the stable isotope signal. Whilst I agree, much of this paper continues to focus on the biota.

Is the methodology sound? Does the work meet the expected standards in your field?

Yes.

More could be done as a second paper, developing the modelling experiments and bringing in more data. But for the current hypothesis, this is enough.

Is there enough detail provided in the methods for the work to be reproduced?

Yes.

Suggested improvements

The changes to the previous version made by the authors have addressed most of my concerns.

Two minor observations/suggestions for this version:

1. Lines 207-208: I suggest adding a connecting sentence explaining why this section in Algeria is important for this study and the modelling. For example, "The Sidi Zidane section.... (Figs 6 and 7). This section provides an opportunity to examine changes in the biota and geochemistry that may be responses to salinity variations investigated through the climate modelling." (This is a suggestion only!)

2. Discussion. This is now a long section. There is much interest and importance here, but I admit I got lost in it once or twice. There is also some repetition. I do not think that it would take much to tighten this section up. My suggestion would be to go through and highlight with a marker the key takeaways - I find using the Read Aloud facility in MS Word quite useful (I am sure you already do this).

Clarity and context

This version is far clearer, and I thank the authors for making the effort to take my comments and those of my fellow reviewers on board.

One typo, (line 425), I think you mean "statistically"?

Reviewer #3

(Remarks to the Author)

The manuscript now reads a lot better than the initial draft and it is easier to follow the authors' line of reasoning.

In the review of the revised manuscript, I focussed mostly on the implementation of my original comments. As such, I am satisfied that my suggestions raised in the annotated pdf of the original manuscript have now been implemented. I attached another annotated pdf with some additional minor suggestions, mostly editorial.

The major comment I had in the previous iteration of the manuscript referred to the handling of the relative abundance data, especially that of the palynological analyses. The authors transferred the compositional diagrams from the supplementary data to the main part of the manuscript (Figure 8A). This now allows the reader to assess the changes of the overall assemblages. The authors also conducted compositional data analysis using a centred log ratio which shows the relationship between *Areoligeraceae* and the ratio of *Guembeltria/Planohedbergella*.

One major comment could not be addressed. No marker grains were added to assess whether dinoflagellate cyst assemblage changes were mainly due to changes in paleoproductivity or changes in sedimentation rate. Because the authors integrate their new data with legacy data, which they have no control over, it is difficult to address this. Having said this, the very significant abundance changes of *Areoligeraceans* combined with the ratio of *Guembeltria/Planohedbergella* alongside the geochemical analyses make it more likely that environmental changes are responsible. I am satisfied that this point has been addressed as well as possible.

I also note that CIE 1 now has been redefined and fits the criteria outlined in the manuscript better than in the original version.

Response to the Editor and Reviewers

We sincerely appreciate the thoughtful and constructive feedback provided by the Reviewers on our manuscript, now titled '*Ocean freshening near the end of the Mesozoic*'. Each comment has been carefully considered, and we have addressed all the points raised by incorporating the necessary revisions into the manuscript. Additionally, we have submitted the improved and new figures as well as additional supplementary materials. We believe these revisions have substantially enhanced the manuscript and, with the Reviewers' valuable insights, it is now in a more refined state, ready for your consideration. We believe that the improvements made will meet the high standards of *Nature Communications* journal.

Regarding the Editor's primary concern about the selection of parameters in the modelling experiments, such as the choice of water depth, we would like to offer a detailed explanation:

One of the Reviewers noted that the water depth of the Panama island arc in the Maastrichtian could be comparable to that of an island arc in the western Pacific today. We agree that this scenario cannot be excluded. This is why one of the water depth values selected for the modelling experiments was 500 m. However, as noted in the revised manuscript and explained in the new Supplementary material S3, the Panama island arc was developed on top of a volcanic (oceanic plateau) promontory and this is the rationale to favour a bathymetry shallower than 500 m. In comparison, the modern Antilles island arc has an average water depth of approximately 200 m even though it did not develop on top of a promontory (a more comprehensive comparison of water depths from various arcs is presented in the new Supplementary material S3).

In addition to the experiment testing the depth of 500 m, our objective was to test marginal 'boundary conditions'. The goal was not to determine the exact depths within the Central American Seaway (CAS), but to explore the potential ocean current reorganisations that could be triggered by changes in the depths of CAS. This was essential for explaining the specific microfossil assemblage we found in the Maastrichtian sediments. Therefore, we have chosen additional CAS depths that render a wider phase space for exploring the limits of the dynamical imprint.

The selection of the deepest conditions (1700 to 2500 m) was supported by paleogeographic data from Markwick and Valdes (2004). The shallowest CAS condition was set to 10 m to ensure unidirectional water mass transport (exclusively from the North Atlantic to the Pacific) preventing the inflow of deeper Pacific waters into the Atlantic. This controlled experimental isolation of the Central American Seaway served as a critical end-member scenario, allowing for a unique assessment of the impacts of Arctic gateway opening and closure on North Atlantic salinity and temperature, independent of Pacific water influence. This approach yielded to be crucial, as this way we established that only the combined effects of CAS shoaling and Arctic Ocean restrictions can drive global changes consistent with our geological proxies.

Though extreme, this depth of 10 m should not be considered as unrealistic. The palynological assemblage from Maastrichtian sediments in Guatemala (CAS) studied by the first author (Radmacher et al., 2021) support this possibility. Although the palynomorphs exhibit low diversity, they are represented by areoligeracean morphotypes typically associated with shallow, inner-neritic zones and high energy regimes (Brinkhuis and Schiøler, 1996; Pross and Brinkhuis, 2005). The inner neritic environment typically refers to the shallowest part of the continental shelf, extending from the shore to a depth of about 50 m. Even shallower depths in CAS are further supported by the occurrence of the Angostura Formation, which represents a carbonate platform that during the Maastrichtian was extended across Guatemala and Mexico (Fourcade et al., 1999). This formation is known of large benthic foraminifera (including alveolinids), which typically live on the seafloor of shallow, tropical and subtropical seas in carbonate-rich environments, such as reefs and lagoons. For comparison, the present water depths of the Bahama carbonate platform are generally very shallow, typically not exceeding 25 meters, with 60 % of the platform (61,400 km²) lying at depths of 5 meters or less (Harris et al., 2014). All the mentioned above data indicate that the general depth of the CAS during the Maastrichtian could be indeed that shallow, and even partially exposed, as shown by Buchs et al. (2010, fig. 9C) on his proposed model for the CAS arc development during Maastrichtian. Some authors even indicate the presence of "land bridge or closely located islands" between Americas that may have been present as early as from the latest Campanian/earliest Maastrichtian (Iturralde-Vinent, 2006; fig. 5b). According to the author, these landforms were shallow enough to facilitate the interchange of terrestrial vertebrates between the both continents (Ortiz-Jaureguizar and Pascual, 2011).

As pointed out by Reviewer 1, none of the tested scenarios could be considered entirely realistic. Our overall goal was not to estimate the exact depth of the CAS, but to observe the trends in paleoenvironmental change driven by shallowing in the CAS in combination with changes in the northern gateway configurations to deduce the optimal fingerprint by the underlying effects in comparison pattern of geological proxies. The resolution of global modelling cannot fully capture features such as islands or underwater volcanoes. However, additional conclusion from our methodological approach is that the average depth of the CAS, characterized by a general uplift and shallowing

trend towards the late Cretaceous (e.g. Buchs et al., 2018; Radmacher et al., 2021), during the Maastrichtian may have ranged between 500 and 10 meters. This range is shallower than previously suggested (e.g. Valdes and Marckwick, 2004).

The conclusion aligns also with recognized trends regarding global ocean shallowing towards the end of Cretaceous, that would intensify the rate of land uplifting in the CAS. The global shallowing is also supported by our palynological record, where Maastrichtian intervals from various regions near the Atlantic Ocean are dominated by Areoligeracean dinoflagellate cysts. In contrast, the early-Late Cretaceous sediments are characterized by a richer and more diversified assemblage of ‘normal’ marine species (e.g. Radmacher et al., 2014a, 2014b, 2014c; 2015; 2020). Areoligeraceae, which appear in exceptionally high-abundances during the Maastrichtian, apart of signalling shallow, inner-neritic environments (Brinkhuis and Schiøler, 1996; Pross and Brinkhuis, 2005), also serve as proxies of increased water stratification (Frieling et al., 2018; Vellekoop et al., 2015).

Subduction of the Pacific Farallon Plate beneath the Caribbean and South American plates, began around ~75-73 Ma, in the Late Cretaceous (Buchs et al., 2010), and the subsequent collision between the Central American Peninsula and South America resulted in continuous uplift in the region, eventually leading to complete land exposure near the beginning of the Quaternary. However, the oceanic changes commenced much earlier, long before the formation of the land bridge, and their precise timing had remained uncertain. What makes our work unique and novel is that based on the model simulations we identified the threshold driving ocean current reorganizations in the Central American Seaway, which influenced global Maastrichtian fossil assemblages long before significant exposure of land. In addition, we found that the combination of CAS shoaling and Arctic Ocean restrictions is necessary to explain our geological proxies. These findings provide crucial insight into the oceanic dynamics that preceded the formation of the land bridge between the Americas.

Thanks to the insightful comments of the Reviewers we recognized that our methodological approaches and experimental setup were not clearly explained in the original manuscript. In response, we have revised and improved all the sections of the manuscript and discussed the reasons for the specific depth selection (Materials and methods section, starting from third paragraph). We believe these changes clarify the structure of our experiments and findings.

Main changes incorporated into the new version of the manuscript:

- The title has been improved by adding ‘Ocean’ at the beginning as suggested by the Reviewer, and by removing the question mark.
- A revised version of the abstract, closely following Reviewer 2’s suggestions has been provided.
- All figures have been improved.
- Additional figures (now Figure 2, 5 and 8) have been provided.
- Figure 3 has been modified to more clearly demonstrate our findings.
- Figure 4 has been replaced (now Fig. 5).
- Additional Supplementary material (S2I to S2III) has been provided to illustrate variations in global water salinity and temperature distribution at different depths, depending on the paleogeographic changes.
- A new Supplementary material (S3) has been added to explain in detail the depth characteristics of the Central American Seaway during the Maastrichtian.
- Last subchapter ‘Maastrichtian Ocean freshening’ has been substantially rewritten and replaced by the new text.

Below we address the Reviewer concerns individually. All these explanations are incorporated into the new version of the manuscript.

Sincerely,

Wiesława Radmacher and co-authors

REVIEWER COMMENTS

Reviewer #1 (Remarks to the Author):

The article by W. Radmacher et al. investigates the freshening of the oceans in the late Mesozoic era, a crucial process in ocean dynamics with significant implications for marine biology. The study focuses on the period just before the K-Pg (Cretaceous–Paleogene) boundary. While previous research has primarily examined the effects of the Chicxulub impact and the Deccan Traps, the role of ancient seaways has largely been overlooked, despite the fact that their evolution likely had substantial impacts on ocean circulation and water mass distribution.

This manuscript seeks to address this gap by clarifying the influence of seaways on oceanic changes during this period. The authors employ climate modeling to analyze the seaways' effects and examine shifts in marine organism populations as indicators of environmental change. The use of these two approaches to support the argument is promising. However, some conclusions lack precision or are not sufficiently supported by the data. In particular, I find that the analysis of the climate simulations is somewhat superficial. In its current form, this study cannot be published. It requires substantial revision, particularly in terms of clarity, precision, and support for its conclusions. The methodology itself is valid, but the Materials and Methods section lacks sufficient detail to allow for the reproduction of the digital experiments. The manuscript lacks precision and specific quantification in places. Additionally, some of the figures do not clearly illustrate the changes discussed in the text.

- We have thoroughly revised the manuscript in accordance with all Reviewer comments. Specifically, we have addressed the main concern raised, as detailed above, and have provided a clearer explanation of the issue of the water depths selected for the experiments. The methodological chapter has been significantly improved to ensure greater clarity and precision. Additionally, we have refined the expression of our findings to present them more accurately. The figures have also been improved for better clarity and visual impact, and we have provided more detailed explanations where necessary. We believe these revisions comprehensively address the feedback and significantly strengthen the manuscript.

My comments:

The paleogeographic framework used in this study is based on Markwick and Valdes (2004), where the Caribbean plate is bordered to the west by an island arc associated with subduction—a chain of islands separated by narrow seaways—and to the east by a second arc. The authors suggest that the shoaling of CAS may have restricted exchanges between the Pacific and Atlantic Oceans. According to studies by Buchs et al. (2018), these low exchange rates potentially contributed to the Oceanic Anoxic Event 3 (OAE3) around 90-84 Ma. The authors propose that the shallow Central American Seaway (CAS), along with progressively restricted marine connections with the Arctic Ocean, limited water exchanges between the Pacific, Atlantic, and western Tethys. The study focuses specifically on the late Maastrichtian period. However, the authors do not clearly specify whether these exchange restrictions, from a geological standpoint, were in effect during this particular time frame.

-From a paleogeographic perspective, the early arc volcanoes began to develop during the Campanian and influenced the Maastrichtian paleogeography. From that time, the area became an island arc, developed on an oceanic plateau basement, which could have shallowed more rapidly than a typical intra-oceanic volcanic arc. We agree that we should refer to the Maastrichtian, since the paleontological and paleogeographical data used in this study covers this time interval. This is now changed and more clearly indicated throughout the manuscript text.

The initial results focus on depth sensitivity experiments for ocean passages, conducted using an Earth system model described in the Materials and Methods section. However, the boundary conditions, such as pCO₂, solar constant, and vegetation, are not specified. Based on Niezgodzki et al. (2019), we can infer that a pCO₂ concentration of 1120 ppm was likely used. The authors test various configurations, including one with the Central American Seaway (CAS) at a depth of just 10 meters. The choice of the 10-meter Central American Seaway (CAS) scenario appears to align with the available data, but the authors do not explain whether such a scenario is realistic. Why was such a shallow depth chosen, and is this realistic given the presence of an island arc? More discussion is needed on this point to assess its plausibility. The second experiment combines a CAS depth of 10 meters with additional restrictions on Arctic Ocean exchanges. According to Niezgodzki et al. (2019), the Arctic passages were already quite shallow (<50 or 100 meters). However, the authors do not clarify

the specific parameters they applied for this restricted Arctic configuration in their new experiment.

-These issues have now been addressed in the Materials and Methods chapter, Numerical model simulations subchapter, starting from paragraph 2.

The authors note that experiments with a 10-meter-deep Central American Seaway (CAS) yield the most pronounced paleoenvironmental changes, particularly in terms of water column stratification in the Arctic Ocean. However, it is unclear how these changes are illustrated in Figure 2C4. In general, Figure 2 is challenging to interpret due to the small size and limited contrast of the color scale, making it difficult to discern the changes described. The authors also mention that restricted exchanges through the Greenland-Norwegian Seaway increase temperature gradients in the Atlantic Ocean and western Tethys. However, this effect is hard to see in Figure 2D2, as the data appear to be globally averaged. The authors tested additional scenarios, with and without the Western Interior Seaway (WIS) and Hudson Seaway (HUS). They argue that the 10-meter CAS configuration best represents conditions at the end of the Cretaceous—specifically, the Maastrichtian.

-The past Fig. 2 (now Fig. 3) is now improved to more clearly indicate our findings. We observe that the most intensive water column salinity stratification in the Atlantic Ocean occurs in the scenario where the CAS is 10 m deep (Fig. 3.2C and 3.2D). However, in the case of open marine connections around the Arctic Ocean (AO) combined with CAS shallowing, surface salinity in the AO increases (Fig. 3.1C), which we consider unrealistic, based on our palynological record showing a dominance of peridinioids in that region, indicating freshening during the latest Cretaceous. Therefore, we conclude that the scenario with the shallowest CAS and restricted marine connections around AO (Fig. 3.D1-4), which leads to a decrease in sea-surface salinity in the Arctic Ocean, is the most realistic. We further explore and illustrate additional findings in Figs 4 and 5.

On what basis is this depth chosen, and does this imply that these conditions were unique to the Maastrichtian?

-The rationale behind the selection of depths used in the numerical simulations is explained in detail above, at the beginning of the document. The conditions we interpreted are indeed specific to the Maastrichtian, as from this interval come our Maastrichtian proxies. The manuscript has been revised to clarify this point.

Additionally, what mechanisms could account for a change in CAS depth by the end of the Cretaceous?

-The shallowing in the CAS can be attributed to formation of an island arc on top of an oceanic plateau promontory, as explained in the new Supplementary materials S3.

In the second part of the study, the authors analyze multiple records, starting with the Sidi Ziane section in Algeria and boreholes in the Norwegian Sea. However, some sections presented in the appendix (e.g., Argentina, USA) are never discussed in the manuscript.

-The Maastrichtian dinoflagellate cyst assemblages in all sections presented in the supplements, including those studied herein and additional available data are dominated either by Areoligeraceae (in low latitudes) or Palaeoperidiniaceae dinoflagellate cyst (in high latitudes). We interpret these findings as indicating increased freshening in the Arctic Ocean and heightened water stratification in the Atlantic. A detailed discussion on the illustrated data and the environmental preferences of the dinoflagellate cyst groups has been further developed in the main text of the manuscript, chapter 'Global dinoflagellate cyst signals from Maastrichtian sections'.

Lines 211-214: The abundance of certain organisms is interpreted as indicative of shifts in temperature or nutrient-rich conditions, potentially driven by upwelling or runoff. The authors conclude that significant environmental changes occurred prior to the K-Pg boundary (KPB). Borehole 6707/10-1, which spans approximately 3 million years (~70–67 Ma), shows relatively minor variations (see Fig. S4B). In borehole 6711/4-U-1, there is a brief increase around 68 Ma; however, this occurs well before the KPB, and fluctuations between 68 Ma and the KPB do not appear markedly different from those preceding the 68 Ma event.

Figure from supplementary material (S1) illustrates the differences between low- and high-latitude assemblages, with Areoligeraceae dominating the Maastrichtian interval in low-latitudes (Algeria, Israel, USA) and Peridiniaceae in high-latitudes (northern North Atlantic and Argentina). This record is important for understanding the latitudinal distribution of dinoflagellate cyst groups and supports the modelling results, which suggests freshening in the Arctic Ocean and water column stratification in the Atlantic. Changes between the Maastrichtian and older assemblages is recognised in previous studies. Increase in Palaeoperidinium pyrophorum and other peridinioids in the uppermost Cretaceous of the Barents Sea is (Radmacher et al., 2014a,

table 1 and 2, and 2014b, figure 3, cores 7119,12-1; 7119/9-1; 7120/7-3; 7120/5-1; 7121/5-1) and domination of *Isabelidium* and *Chatangeilla* spp. in the Norwegian Sea (Radmacher et al., 2015, fig. 4, cores 6707/10-1 and 6711/4-U-1 suggests an increase in freshening in the northern North Atlantic towards the end of the Cretaceous.

Lines 236-244: In the Discussion section, the authors suggest that a shallow Central American Seaway (CAS) (10 meters) increases water mass stratification in the Atlantic Ocean and western Tethys. However, there is no geological evidence supporting a continuous shallow barrier between North and South America; instead, geological data indicate the presence of an island arc associated with the subduction zone.

-The shallowing of the CAS has been thoroughly addressed at the beginning of this document and in the Materials and Methods section of the main text. Broader regional considerations regarding shallow depths in the CAS are also provided in the newly added Supplementary Material S3.

Lines 246-256: The authors propose that increased stratification occurred in the Atlantic Ocean and parts of the Tethys, yet they do not illustrate these changes in the figures. To support this claim, visual evidence of stratification changes should be provided. Additionally, the mechanisms underlying these changes—particularly the interactions between the CAS and Western Tethys—are not clearly presented or discussed.

-The visual evidence of stratification is provided in the new Figure 5. Concerning the simulated region, we have now provided explanation stated in the text and shown in the Fig. 1 that the analysed region includes both the Atlantic Ocean and Western Tethys (Region A).

Lines 261-266: The authors suggest comparing the scenario-induced changes (CAS uplift) with geological data and note that these processes coincide with global cooling and sea-level fall. However, the timeframe remains unclear. What specific period are the authors addressing? Based on palaeontological data, this study focuses on the Maastrichtian, a time marked by warming followed by a brief cooling just before the K-Pg boundary. The Earth has experienced global cooling since the Turonian, but the late Maastrichtian includes a notable temperature rise prior to this cooling episode.

-We agree with the comment. The micropaleontological data used in this study indeed cover the Maastrichtian interval, and this has been clarified in the main text. The new version of the revised manuscript focuses primarily on paleogeographically driven global salinity budget changes rather than temperature variations.

Lines 291-343: The authors suggest salinity changes in the Sidi Ziane section, possibly linked to increased continental runoff (L310). For the Norwegian Sea sections, they similarly conclude that runoff may have increased, contributing to water mass stratification (L341). The authors propose that these salinity changes result from shifts in ocean circulation driven by evolving ocean passages. While the Norwegian Sea sections indicate a salinity response (although the figure is difficult to interpret), the authors do not clearly present salinity changes in the Tethys from their sensitivity experiments. Additionally, changes in salinity can also stem from shifts in the local precipitation-minus-evaporation (P-E) balance. Modifying ocean circulation could potentially alter atmospheric circulation as well—especially given the possible removal of the Western Interior Seaway (WIS) and Hudson Seaway (HUS)—which could impact the P-E balance. However, this point is not addressed.

-As explained in the new version of the manuscript, the simulated region in Fig. 3 includes Atlantic Ocean + Western Tethys (see the Region A in Fig. 1).

-To address concerns regarding the P-E balance, we have prepared an additional figure (see below). This figure is now also added as Supplementary Data S2B. It presents simulated P-E in the North Atlantic-Arctic region under different gateway configurations between the Arctic and proto-North Atlantic Oceans, as well as varying CAS depths. The results indicate that changes in P-E between simulations are minor and confined only to local areas. Therefore, it is very unlikely that salinity changes in the North Atlantic region were driven by P-E changes.

The authors discuss the roles of the Deccan Traps and the Chicxulub impact, suggesting that these events may have masked the effects of ocean passage changes. The establishment of the Deccan Traps and the associated CO₂ emissions likely increased pCO₂ in the late Maastrichtian, leading to global warming, which could plausibly increase continental runoff. However, implicating the Chicxulub impact as a contributing factor seems more challenging, given that it occurred precisely at the K-Pg boundary, after the Maastrichtian.

-When we refer to how Chicxulub's impact could have masked the signal, we are thinking of the early Danian since due to the catastrophic mass extinction event sudden global climatic and environmental perturbations could have masked shifts in ocean circulation or more precise paleoceanographic reconstructions.

While the authors emphasize the impact of ocean passage changes, there is insufficient geological evidence and temporal constraints to substantiate a causal link between these geological and oceanographic events. Why would the CAS have been shallow during or prior to the Maastrichtian? While some studies suggest a southern closure of the WIS, the northern portion of the WIS, along with much of the HUS, appears to remain open until the K-Pg boundary.

The geological evidence, as discussed above, indicates shallowing conditions in the Central American Seaway during the Late Cretaceous, a conclusion also supported by Buchs et al. (2018) and Radmacher et al. (2021). This information formed the basis for conducting numerical experiments to investigate paleoceanographic changes, including the reorganisation of oceanic circulation driven by restricted connections between the Atlantic and Pacific Oceans. Regarding the WIS and HUS, we simulated restrictions in the connections between the Arctic Ocean and the rest of the Global Ocean, resulting in limited exchange of water masses. Even the southern closure of the WIS could have significant impact on water mass exchange and increased outflow through Greenland-Norwegian seaway. Given the uncertainty surrounding the paleogeographic conditions during the latest Cretaceous, the goal of our experiments was to compare the results with available geological proxies. Following the manuscript revision, we now discuss these comparisons in greater detail, not only with the presented Maastrichtian data but also in relation to boarder geological evidence, including the

palynological record from the northern North Atlantic (Radmacher et al., 2014a, 2014b, 2015; 2021). We believe that this revision addresses the Reviewers' comments and strengthens our interpretations. Variations in marine connections around the ArO are based on published data, e.g. Ladant et al. (2020) and Schröder-Adams (2014).

Reviewer #2 (Remarks to the Author):

Manuscript# NCOMMS-24-63460-T

Corresponding Author: Wieslawa Marta Radmacher

Title: Freshening near the end of the Mesozoic?

Reviewer Assessment

Key Results

The key results from the paper, as I understand them, are that the model experiments show that shallowing gateways result in an enhanced freshening of the oceans in the Maastrichtian. This is similar to what has been found in studies of the Cenomanian and Early Eocene. This would suggest that such changes to the ocean system may be more common than currently apparent in the literature.

Validity

The climate model experiments show ocean freshening (lower salinity) with increased isolation of ocean basins. This is neither new nor a surprise. Nonetheless, I believe this study to be a useful addition, but I am concerned by the following:

1. Setup of the paper – I re-read the manuscript many times and feel that the abstract and introduction would both benefit from being ‘tightened up’. By this I mean the following:

(a) A more explicit statement of the aim of the study – is the aim to show how paleogeography affects the potential for freshening of the oceans in the geological past. That was my take.

(b) What then confused me as a reader were the statements about the importance of understanding climate change, the causes of extinction, paleoecology, etc. Ok, yes, but these are not fully developed, and they feel like they have been added to the abstract and introduction to capture all the current buzz phrases. Yes, the potential for freshening of the oceans and its impact on ocean ecology and circulation is important. However, this study is about longer-term factors (viz. paleogeography), whereas today, we are concerned with short-term drivers, like CO₂, land use changes and how these impact the hydrological system.

(c) Extinction – there is mention of the study interval preceding the mass extinction at the end of the Cretaceous (lines 33-35, 62-64). But a relationship between environmental changes (as described in this paper) and extinction is not then developed. I cannot see that they are related except for certain marine species being affected by salinity changes, if these are global, which does not seem likely. Is the intention to state that there are environmental changes leading up to the end Cretaceous which have not been examined enough?

-We acknowledge these concerns. The abstract has been revised to clearly state the aim of the study. In the updated version of the manuscript, we have addressed all the issues raised and implemented changes to better articulate our objectives and findings. As the study focuses on long-term changes, the previously mentioned statements have been either removed or modified accordingly. Overall, the manuscript has undergone substantial revisions, and we have made every effort to eliminate the ambiguities highlighted above by the Reviewer.

(d) Abstract – this needs to be more formulaic: what I would do is to start by (sentence 1) stating that the study is aimed at understanding how common ocean freshening might be in the geological record (if that is indeed the aim); (sentence 2) How this might be important for future changes to the Earth ; (sentences 3 and 4) that the study does this by (1) running modelling experiments (long-term changes) and (2) outcrop sites to test model results in those areas; (5) That the results indicate that freshening may have occurred in the Maastrichtian.

-We are grateful for these helpful suggestions. We have reorganised the abstract strictly following the recommendations of the Reviewer.

2. Modelling experiments - The validity of using a 10m bathymetric experiment for the Central American Seaway and using this to make conclusions. Regarding climate modelling, a 10m water depth is essentially land! It is also unlikely to be the case for the entire length of the arcs that form the barriers. The authors could do no worse than look at the present-day bathymetries of the island and continental arcs in the western Pacific. Especially the discontinuous nature of some of these. Do I think that the Central American Seaway has barriers to deep water flow based on the geology I know? Certainly! I, therefore, agree with the authors. But, I do not consider this new, although it is useful.

The primary objective of this study was to determine the depth threshold of the Central American Seaway (CAS) that would trigger a reorganization of water circulation, potentially influencing the latest Cretaceous micropaleontological assemblages based on the available geological record. Our findings indicate that this threshold falls between 500 and 10 meters, with geological evidence suggesting it was closer to 10 meters. To simulate a unidirectional flow of oceanic waters from the North Atlantic to the Pacific—preventing the intrusion of deeper Pacific waters into the Atlantic—the shallowest CAS condition had to be set at 10 meters. This approach was crucial for isolating the effects of Arctic gateway opening and closure on North Atlantic salinity and temperature, independent of Pacific water influence. Although a 10 meter depth may appear extreme, it should not be considered inherently unrealistic, as geological evidence supports the occurrence of similarly shallow conditions. A detailed discussion of these evidences as well as of the reasons for selecting the specific CAS depths is provided on the first pages of this document, as well as in the main text within the Materials and Methods section under Numerical Model Simulations (beginning with the third paragraph).

3. The outcrop data – this section is fine and, to me, is a paper in its own right, especially the Algerian site. However, for the general aim of the paper, if I understand this correctly, I am concerned that all the localities used in the study to indicate freshening of the oceans are near continental areas and are therefore much more susceptible to changes in freshwater fluxes. They need not indicate global changes (though I think this is what the authors hope through the isotope data? If so, this needs to be drawn out more clearly).

-Our purpose with the isotope data, rather than establishing global interpretations, was more focused on age model construction since we tie $\delta^{13}\text{C}$ curves between Algeria and the astronomically tuned section of Zumaia, Spain. The $\delta^{13}\text{C}$ based correlation is a reliable approach, as demonstrated in several Maastrichtian sections across different latitudes and environments (Batenburg et al., 2018). This way we further refined the age model instead of only extrapolating between the available biostratigraphic datums. Moreover, we used isotopic data in combination with foraminiferal, dinocyst, and CaCO_3 data for the paleoenvironmental interpretation of Sidi Ziane. The palynological signal indicates general significant changes that occurred during latest Cretaceous. This include both shallowing and freshening in selected regions. We learn from global modelling experiments that shallowing contributing to oceanic restrictions could force significant environmental reorganisations. We discuss the modelled scenarios and compare the results to local geological proxies. We believe that the concern mentioned in the comment above is now widely addressed within the significantly improved new version of the manuscript.

4. The equation of ocean freshening and stratification (e.g. line 343). Although freshwater fluxes to the ocean can and do lead to water column stratification, in other circumstances, they can result in instability and overturning.

-We agree with that statement.

5. "...the effect of salinity is often speculated or even overlooked" (line 334). Is this really true? There are numerous papers on density stratification and the role of salinity on circulation, going back to at least 1982 and Brass and 1998 and Hay et al. More recently, this has been discussed by Ladant et al. (2020), which the authors include in the reference list.

-We have improved this part of the text accordingly.

Significance

The conclusion that paleogeography may be important for restricting ocean basins and enhancing the potential for 'freshening of the ocean in these basins' is not new, but it is useful. The important role of gateways on paleoclimate has long been known (see Berggren 1982 as a start). That the authors identify this for the Maastrichtian to add to previously published work on the Cenomanian and Early Eocene is, to me, a useful addition. However, I am less clear on how the outcrop data fits into testing this, given my concern (see above)

that the localities are all near-shore and susceptible to local hydrological changes. This might be an area where the authors could strengthen the arguments using the same data.

-This issue has been more clearly addressed in the text, following the advice of the Reviewer.

Data and Methodology

Including observational data and model experiments is a common and powerful approach for examining the response of the Earth system to changes in boundary conditions. To that extent, the data and methodologies are valid. However, I am struggling with the detailed analysis of the Algeria site data and the bigger-picture question being discussed. This may be me, but there feels like a disjunct here. It is possible that this could be addressed by rewriting the section or turning the whole paper around, viz., “how the analysis of the Algeria data suggests changes to the ocean system and hydrology, that are then examined using the modelling”, rather than the modelling followed by the data. This is just a thought. I am less experienced with biostratigraphy and palynology, but from what I do understand, this all seems valid.

-Thank you very much for this valuable comment. It would indeed be preferable to use this approach, but only if we had a complete geological record covering not only the Maastrichtian but also the Campanian and even Santonian. Unfortunately, the uppermost Cretaceous record is challenging, and we have determined that the modelling results provide a more complete picture. This is why we discuss it at the outset. However, in the new version of the text the Sidi Ziane section is discussed and compared with other sections worldwide, allowing for a more detailed interpretation and comparison with the modelling results.

The modelling relies on one published Maastrichtian paleogeography. What was the rationale for using this one? There is an updated version to this map if the authors are interested.

-The detailed explanation of used paleogeographical constrain and selected water depths used in our simulations is in detail explained at the beginning of this document, and within the updated manuscript text.

Analytical approach

The analysis of the biostratigraphy data is outside of my principal area of expertise.

Suggested improvements

My recommendation from reading this manuscript is that it would greatly benefit from a more explicit statement of the project aims and objectives and a clearer link between the aims of the Algeria analysis and the big-picture modelling study.

Clarity and context

1. Although the editorial guidelines say not to focus on grammar, I would urge the authors to avoid using words and statements such as the following:

- (a) “favouring” (line 35) – I would be more direct, for example “indicate”, “show”.
- (b) “Seem to” (line 39) – instead use “...processes may have played...”
- (c) “We assume that” (line 152) – either your data shows this, or your assessment of the existing literature shows this.
- (d) “Reliable” (line 154) – what do you mean by “reliable”? Is this the view supported by data, or the most recent publication, or something else?
- (e) “Novel” (line 230; but used several times in the manuscript) – In this case, this is not a novel (meaning “new”) mechanism. “Novel” should only be used if the mechanism has not been recognized previously.

-All the issues suggested above have been resolved.

2. Title: I would add the word “Ocean” at the beginning of the title to make the context of “freshening” clearer to readers.

-We agree with this comment and we have changed the title accordingly.

3. When you use “Open-marine” (Line 113), do you mean deep-water connectivity? The system is open to the

global ocean system, just at shallow depths. For example, the Atlantic and Pacific were not separated in the Late Cretaceous as they are today by Central America.

-By 'open-marine' we refer to oceanic conditions in which the connections between the Arctic Ocean and the rest of the global ocean were not restricted, allowing for uninterrupted water flow.

References

There is a comprehensive reference list, although there are a few papers I think the authors should have a look at, for example, Laugié et al. (2021), <https://agupubs.onlinelibrary.wiley.com/doi/full/10.1029/2020PA004202>

-This reference has been included in the new version of the manuscript. Laugié et al. (2021), by testing different depths in the Central American Seaway (CAS) referring to the Cenomanian (4000, 2500, and 300 m), conclude that shallower conditions lead to the development of complete anoxia in the Central Atlantic basin. This experiment was conducted within a different time interval, and the applied water depths varied. However, the overall conclusion is that the depth of the CAS plays a crucial role in the reorganization of water circulation, influencing the distribution of oceanic conditions and potentially driving significant biogeochemical changes. This corresponds well with our results.

Overview

This manuscript describes some very interesting results, and there is much that I think would be useful to see published. However, as I have described above, I think the paper needs some reorganization and further thought. I encourage the authors to make changes, and I wish them luck in their research.

Reviewer #3 (Remarks to the Author):

This study integrates modelling results with micropaleontological and geochemical data from four geological sections in a time slice near the end of the Mesozoic (around 66Ma), just preceding the K-T extinction event. Using the COSMOS Earth System Model, Radmacher and co-authors test five scenarios with varying depths of the Central American Seaway (CAS) and marine connectivity around the Arctic. By shoaling the CAS, the model shows an increased salinity stratification in the Atlantic, an effect that is exacerbated when restricting seaway connections around the Arctic. The authors compare the modelling results with previously published micropaleontological and geochemical data (supplemented with additional analyses) from four localities: the Sidi Ziane Section (Algeria), the Re-2 Core from the Negev region (Israel) and two Norwegian Sea cores. They report increased relative abundances of dinoflagellate cysts belonging to the Family Areoligeraceae which is surmised to be a result of increased stratification. Increased abundances of protoperidiniaceans, known heterotrophs, are interpreted as evidence for the presence of nutrient-rich water prone to salinity fluctuations. Taken together, the authors conclude that their modelling and data point to a freshening at high latitudes with an increased outflow of low salinity waters through the Greenland-Norwegian Seaway.

The authors should be commended because they present results from a united effort of modelers and data driven scientists in a single study which will therefore be of interest to the diverse readership of Nature Communications. I find the manuscript easy to follow, but found a number of grammatical and formatting issues listed as comments in the annotated pdf (attached). The text is accompanied by clear figures that effectively convey the authors' message.

-We thank the Reviewer for the comments. All the issues outlined in the attached pdf file have been addressed.

Major Comment

Relative abundances are a form of compositional data (their total summing to 1) meaning that a rise in one species is automatically at the expense of another (the constant-sum constraint). In figure 6, selected profiles of planktic foraminifera and dinoflagellate cysts are presented which are difficult to interpret without knowledge of the whole assemblage composition (which the authors nicely present in the figures of the supplementary data). This leads to interpretations that are not necessarily backed up by the data. For example, on Line 294–295 the authors claim “when Guembelitra is more abundant than Planohedbergella, Areoligeraceae show higher abundances”. This is clearly not the case in CIE1 where areoligeraceans are close to 100% of the palynological assemblages while relative abundances of Planohedbergella exceed those of Guembelitra. Such covariance as

suggested as suggested by the authors on lines 294–295 can only be examined using rigorous statistics that obey the laws of compositional data analysis.

While the original relative abundance data allowed us to infer a direct relationship between *Areoligeraceae*, *Guembelitra*, and *Planohedbergella* based on their respective abundance curves, we agree with the Reviewer who correctly pointed out that this relationship was not supported by statistical analyses or composition data analyses. Therefore, we followed the suggestion and explored the relationship between *Planohedbergella*, *Guembelitra*, and *Areoligeraceae* in more detail. Before doing so, we performed multivariate analyses, specifically Principal Component Analysis (PCA), to robustly establish the main groups we would later discuss. The PCA results showed that PC1 explained 75.903% of the variance, while PC2 accounted for 17.02%, together explaining 93% of the total variance. The genera with the greatest loadings in PC1 were *Heterohelix* (0.756), *Planohedbergella* (-0.534), and *Guembelitra* (-0.364). In PC2, *Guembelitra* had a loading of -0.791, while *Planohedbergella* had a loading of 0.594, suggesting potentially inverse ecological preferences. Loading for *Heterohelix* were irrelevant in PC2. The remaining taxa exhibited negligible loadings in both PC1 and PC2.

Given these results and the better-known ecological preferences of *Guembelitra* and *Planohedbergella* we focused our discussion on these both taxa. Subsequently, we conducted a composition data analysis to accurately compare them with the major dinocyst group, *Areoligeraceae* (See new Fig. 8). We applied centered log-ratio (CLR) transformations to the compositional data to avoid redundancy in relative abundance comparisons. This analysis revealed that *Areoligeraceae* exhibited significantly higher values when *Guembelitra* was more abundant than *Planohedbergella* (Fig. 8C). Additionally, we performed a Wilcoxon test to assess this relationship using the transformed compositional data, resulting in a p-value < 0.05 (0.02371), providing statistical support for our interpretation. We also applied the same approach using the relative abundance data, which remained significant, yielding a p-value of less than 0.001.

In addition, relying on changes in relative abundance, especially for dinoflagellate cysts, without knowledge of the burial flux or concentration of cysts can be challenging. Dinoflagellate cyst burial flux calculations are recommended (they can be calculated if the marker grain method has been applied during preparation) as they can separate rises of number of cysts that are result of increased incorporation of cysts in the sediment from

those that are the result of changes in the sedimentation rate. If these data are available, they would be invaluable in interpreting the presented assemblage changes.

-This is an important suggestion, and we agree with the recommendations. However, the Lycopodium marker has not been introduced for the new samples from the Sidi Ziane and Negev regions, and as such, we are unable to calculate the dinoflagellate burial fluxes. Nonetheless, the key observation from these two sections is that the Maastrichtian intervals are overwhelmingly dominated by Areoligeraceae. This signal is associated with environmental stress, such as increased freshwater input and changes in surface-water salinities, which likely contributed to enhanced water stratification. We now also compare dinoflagellate cyst assemblages from older intervals, and this is based on published data. Furthermore, the lateral distribution of the studied sections and the dominance of different groups at low and high latitudes are important factors, indicating different environmental conditions in these distal regions. In the new version of the manuscript, we discuss all of these findings in detail, providing a clearer and more advanced discussion of the results and their significance.

Other comments

Abstract: why is there no mention of any of the geochemical and micropaleontological data in the abstract? The manuscript discusses an integration of modelling and data but the abstract only mentions modelling results. I believe this is a particular strength of the manuscript and should be included here.

-In the new version of the manuscript, the abstract has been completely reorganized in accordance with Reviewer 2's suggestions. We shortly highlight the use of palynological, micropaleontological, and geochemical data to provide a more comprehensive interpretation of the modelling results. The length of the abstract is restricted, which unfortunately hampers additional detailed explanations.

Line 54: Reference 21 deals with the late Neogene and offers no evidence about the latest Cretaceous as the authors seem to suggest here.

-We cite this manuscript to highlight the differing views on the timing of the restriction of water mass exchange between the Atlantic and Pacific Oceans, with some authors suggesting ~70 Ma, while others indicate 6.2 to 1.8 Ma.

Line 196 and Figure 6A: CIE1 (high $\delta^{13}\text{C}$ corresponding to low $\delta^{18}\text{O}$ and peak in Areoligeraceae) behaves differently than the other CIEs where lower than expected $\delta^{13}\text{C}$ are recorded. This is not explained in the text. Later in the text these CIEs are ignored even though they were deemed important enough to highlight here and in Figure 5.

-After careful examination, we found that CIE1 could be redefined. The actual CIE interval encompasses a longer interval while it should span from 66.44 to 66.40 Ma, with the peak of Areoligeraceae located in the center. As the Reviewer pointed out, the relative abundance within CIE1 does not allow for a clear observation of the relationship between Guembelitria, Planohedbergella, and Areoligeraceae. However, the compositional data analyses indicate that this relationship is indeed well established. This anomaly within CIE1 may be related to other issues related to the limitations of the geological record affecting the paleontological content of the samples or could be influenced by other environmental variables that we do not fully catch on this study.

Line 268: what about preservation of calcium carbonate in cooler waters? This seems like it could be an issue near the bottom of the Sidi Ziane section where %CaCO₃ is very low. Will this not impact foraminifer preservation and the associated stable isotope signal?

-Regarding to the preservation state of the planktic foraminiferal tests in Sidi Ziane, it varies between moderate and good throughout the entire section. No clear relationship has been observed between %CaCO₃ and state of foraminiferal preservation under microscope. In particular, no deterioration in the preservation of the foraminiferal tests of Sidi Ziane has been observed at samples with low %CaCO₃ value. As stated within the text, "The relationship of $\delta^{13}\text{C}$ bulk with $\delta^{18}\text{O}$ bulk shows a nonlinear correlation, $r = -0.159$, $p(a) > 0.05$ thus suggests that the primary trend is preserved".

Line 287: "this group can be considered as indicative of stressed near shore marine environment, with notably fresh-water input and changes in surface-water salinities followed by enhanced water stratification." This seems like an unrealistically specific condition for increased abundance of areoligeraceans. I agree the data shows

raised abundances could be indicative of “stressed” near shore marine environments but “notably fresh-water input and changes in surface-water salinities followed by enhanced water stratification” seems like an overreach. It may be the paleoenvironment suggested in the current study, but I think this must be rephrased as not to sound that an increase in areoligeraceans is automatically indicative of only this specific situation.

-We agree with the Reviewer that this part needed rephrasing. This interpretation was influenced by overall results, including insights from the modelling. This sentence has been now removed, and the interpretations have been improved accordingly.

Lines 292–297: I find this very speculative, and this is definitively not the case for CIE1 were abundances of *Planohedbergella* are higher than *Guembelitra* in the same sample when Areoligeraceans form close to 100% of the dinoflagellate cyst assemblage (Figure 6 D–E). An inference like the one made by the authors in the sentence starting at line 293 (Areoligeraceae showing an inverse paleoecological signal with respect to *Planohedbergella*...), should be backed up by multivariate statistics.

This issue has already been addressed by the earlier mentioned comment.

Line 302–303: higher productivity could indeed result in higher CaCO₃ but post-deposition dissolution is an important factor in this, especially in cooler high latitude water (e.g. Henrich et al., 2002). Again, calculating a dinoflagellate cyst burial flux would give a much more reliable estimate of the organic matter burial rate and therefore productivity. Line 309: “*Guembelitra* bloom”. A high relative abundance is not evidence of a bloom which suggests a high burial flux (high productivity).

*-We consider an increase in the relative abundance of opportunistic *Guembelitra* to more than 20 or 30% to be anomalous and actually evidence of a *Guembelitra* bloom, regardless of how long the bloom lasts. It would not be evidence of a bloom only if this increase in abundance was due to the disappearance/extinction of the other genera with which it is compared in the quantitative analysis (as occurs immediately after the KPB mass extinction, where *Guembelitra* comprises 100% of the planktic foraminiferal assemblages). On the other hand, when a eutrophic taxon as *Guembelitra* increases its abundance with respect to the other non-eutrophic ones, it is indicative of a bloom most likely caused by an increase in productivity.*

Line 327–328: Reference needs replacing with reference number (Harris and Tocher, 2023).

-This reference has been replaced with the number, as suggested.

Line 336–337: sentence needs rephrasing.

-This part has been improved.

Line 359: I agree with the authors here. Only relative paleo-salinity changes can be derived since the exact composition of the Maastrichtian seawater is not known. These relative paleo-salinities can be derived from planktonic foraminifer oxygen isotope records corrected for global ice volume using a normalised global compilation of benthic oxygen isotope records. This has been done for the Pliocene–Quaternary but I am not aware of an equivalent for the Maastrichtian. If available, such a calculation could help solidify the inferred freshening. As the authors point out (e.g., line 396), the studied time interval is ice-free and a conversion of the planktonic oxygen isotope signal using the Shackleton (1974) equation could yield a record that is mainly influenced by salinity changes. Maybe this can be used to give a rough idea of salinity changes?

-Although we find this Reviewer's suggestion very interesting, we lack isotopic analyses on individual species of planktic foraminifera, so we cannot calculate it. Furthermore, we recognise that, despite all its virtues, the the Sidi Ziane section has its limitations regarding planktic foraminiferal preservation, as they do not present pristine tests that are the best suited for single-species isotopic analyses.

Figure 2: check the layering of the red rectangle. Some of the differences discussed in the text are hard to discern at the scale of the individual maps. Add dimensions to the colour gradient variables along the X axis.

Figure 2: “Caribbean Seaway” at the top of the figure is the same as “Central American Seaway” in the caption and text?

Figure 3: add dimensions to the colour gradient variable displayed along the bottom of each figure. HS = Hudson Seaway (correct?). Note that in line 114 the Hudson Seaway was abbreviated HUS instead of HS. Make uniform.

Figure 3A corresponds to the same model parameters employed in Figure 2A (Deep CAS with open Arctic marine connections) while Figure 3B corresponds to Figure 2D (Shallow CAS with reduced Arctic marine connections).

Figure 4A shows the difference between Open GNS/shallow CAS Vs. Open GNS, WIS and HUS/shallow CAS. Figure 4B shows the difference between Open GNS/shallow CAS Vs. Open GNS, WIS and HUS/Deep CAS.

-We greatly appreciate all the valuable suggestions regarding the figures. All of them have been carefully considered, and the figures have been revised and improved accordingly. Additionally, we have incorporated further changes to enhance their clarity and appropriateness. We also added new figures to better explain our findings. Figure 4 has been exchanged.

All the references cited above can be found in the main text of the manuscript.

Whether the data provide unequivocal evidence supporting the posed hypotheses is another matter. However, the only way anyone could address this problem would be to generate planktonic biome maps for the Atlantic using the deep-sea drilling data. However, I concede that correlating these data would not be easy with the fine temporal resolution required. More could be done as a second paper, developing the modelling experiments and bringing in more data. But for the current hypothesis, this is enough.

We are grateful for this thoughtful suggestion. We fully agree that this approach could be highly valuable in addressing additional questions and would indeed be a promising direction for future research. Given the scope and focus of the current manuscript, incorporating this analysis would require substantial time and effort beyond what is feasible here. We therefore consider it an excellent idea to pursue in a dedicated future study.

Discussion. This is now a long section. There is much interest and importance here, but I admit I got lost in it once or twice. There is also some repetition. I do not think that it would take much to tighten this section up. My suggestion would be to go through and highlight with a marker the key takeaways - I find using the Read Aloud facility in MS Word quite useful (I am sure you already do this).

We did our best to implement the suggested changes, while carefully considering the clarity and intent of the original sentences. Although further shortening proved challenging without compromising important nuances, we have made several modifications where possible to improve conciseness and readability.